# Validation of climate mitigation pathways

Pascal Weigmann[1,*], Rahel Mandaroux[1,*], Fabrice Lecuyer[1], Anne Merfort[1], Tabea Dorndorf[1], Johanna Hoppe[1], Jarusch Muessel[1], Robert Pietzcker[1], Oliver Richters[1], Lavinia Baumstark[1], Elmar Kriegler[1], Nico Bauer[1], Falk Benke[1], Chen Chris Gong[1], and Gunnar Luderer[1]

[1]Potsdam Institute for Climate Impact Research (PIK), Member of the Leibniz Association, Potsdam, Germany
[*]These authors contributed equally to this work.

**Correspondence:** Pascal Weigmann (pascal.weigmann@pik-potsdam.de) and Rahel Mandaroux (rahel.mandaroux@pik-potsdam.de)

**Abstract.** Integrated assessment models (IAMs) are crucial for climate policymaking, offering climate mitigation scenarios and contributing to IPCC assessments. However, IAMs face criticism for lack of transparency and limited ability to represent recent technology diffusion and dynamics. We introduce the Potsdam Integrated Assessment Modeling validation tool, *piamValidation*, an open-source R package for validating IAM scenarios. The *piamValidation* tool enables systematic comparisons of variables from extensive IAM datasets against historical data and feasibility bounds or across scenarios and models. This functionality is particularly valuable for harmonizing scenarios across multiple IAMs. Moreover, the tool facilitates the systematic comparison of near-term technology dynamics with external observational data, including historical trends, near-term developments, and empirical findings. We apply the tool in two application cases. First, to scenarios from the Network for Greening the Financial System (NGFS) to demonstrate its general applicability, and second to the integrated assessment model REMIND for near-term technology trend validation, illustrating its potential to enhance the transparency and reliability of IAMs. We apply the tool in two application cases. First, to scenarios from the Network for Greening the Financial System (NGFS) to demonstrate its general applicability, and second to the integrated assessment model REMIND for near-term technology trend validation, illustrating its potential to enhance the transparency and reliability of IAMs.

## 1 Introduction

Integrated assessment models (IAMs) play a prominent role in providing science-based energy assessments and climate policy advice. Early IAM applications date back to the early 1990s (Cointe et al., 2019) and evolved towards the formulation of mitigation targets and the monitoring of political ambition (Van Beek et al., 2020). IAM scenarios are also an important pillar of the assessments conducted by the Intergovernmental Panel on Climate Change (IPCC), in particular with regard to transformation pathways towards achieving climate policy goals. Therefore, they are a central component of climate change mitigation-oriented policies. Edenhofer and Minx (2014) argue that the Summary for Policymakers (SPM) established a collaborative environment where political discussions link with relevant scientific material. They conceptualize scientists as mapmakers in the territory of climate policy. In this interpretation, the role of the Conference of the Parties (COP) can be understood through a metaphor, "the COP operates as navigator, navigating a terrain charted by the IPCC" (Beck and Oomen, 2021, p. 172). In AR6

WGIII Chapter 3, advances in climate modelling are assessed by examining the capability of models to accurately simulate historical developments, underscoring the growing importance of scenario validation in IPCC processes.

Considering the prominent role of IAMs in the climate science-policy interface, IAMs have faced increasing scrutiny. Particularly in policy advice, central criticism ranges from shortcomings regarding model representation (e.g., heterogeneous actors and capital markets) to their ability to capture technology diffusion and dynamics (Keppo et al., 2021). On the one hand, global and national models have often underestimated the rapid technological change in renewable power and demand electrification technologies: for instance, the cost of solar electricity and battery storage has declined by almost 90% (Creutzig et al., 2023) and renewable electricity generation in Asia has more than doubled (IRENA, 2024) in the last decade, far exceeding model assumptions and results. National models face the additional challenge of making assumptions about technology innovation and cost declines, which are largely driven by global developments and trends. Accurately capturing recent trends in technology capital costs is therefore crucial, as they play a predominant role in the sensitivity of model outputs (Giannousakis et al., 2021). On the other hand, near-term IAM scenario results exceeded the rate at which economies are transitioning away from fossil fuels: for example, due to factors beyond pure market-based energy economics, coal power capacities are still being built in some regions despite the declining renewable energy costs. In particular, China and India alone account for 86 percent of worldwide coal power capacity under development (Carbon Brief, 2024).

Not only techno-economic parameters but also the technology representations more generally impact technology trends in IAM scenarios and vary strongly between IAMs, as illustrated by Krey et al. (2019) for the case of electricity generation. They first highlight the need for caution in harmonizing techno-economic assumptions, as they must fit model-specific technology representations and associated projection methods. Second, Krey et al. (2019) participate in the transparency debate (Stanton et al., 2009) by urging the IAM community to publish and openly discuss techno-economic parameters, their definitions, and model technology representations.

This paper introduces *piamValidation*, the Potsdam Integrated Assessment Modeling validation tool for IAMs, to foster its broad adoption in the IAM community and enhance transparency and reliability.

Building on the evaluation framework of Wilson et al. (2021), the *piamValidation* tool facilitates evaluation of historical simulations, near-term empirical validation, stylized fact checks, and model intercomparison, thereby it can be used to contribute to the appropriateness, interpretability, relevance, and especially the credibility of IAM results. This tool and the related input data are meant to be a community resource, where the distribution of the *piamValidation* configuration files can serve as a knowledge-sharing platform and as a performance metric, offering continuous feedback on modeling progress.

In section 2, we present the *piamValidation* open-source R package. The tool enables users to systematically compare variables of large IAM datasets with historical data and feasibility bounds or across scenarios and models. The core function of the tool is to validate IAM results and identify discrepancies. In addition, it also supports harmonizing national and global IAMs in the short and medium term, particularly for current policy scenarios that capture implemented government policies. Furthermore, the tool allows for systematic comparisons of near-term technology dynamics with external observational data as historical, near-term trends or empirical estimates. The *piamValidation* tool is particularly user-friendly and requires only a single command to generate the full HTML report featuring validation heat maps.

In section 3, we present two complementary application cases of the *piamValidation* tool. The first, based on scenarios of the Network for Greening the Financial System (NGFS), illustrates its general applicability across diverse contexts, with an emphasis on demonstrating application types rather than results. The second applies *piamValidation* to evaluate and strengthen the near-term realism of technology trends in the IAM REMIND (REgional Model of Investment and Development, see Baumstark et al. (2021) for details). In the REMIND case study we show how the *piamValidation* tool can be used to first spot the deviations to historical and outlook data and then demonstrate performance improvements near-term realism focusing on offshore wind capacity, carbon management, and electric cars. The rationale for choosing these technologies is twofold: first, the feasibility of their near-term scaling has severe implications for scenarios with high ambition for emissions reduction (Brutschin et al., 2021; Bertram et al., 2024); second, the *piamValidation* tool highlighted relatively large deviations in these variables between the REMIND scenarios and historical data as well as current technology trends. We then present strategies that have been implemented to improve REMIND, and we demonstrate that subsequent scenarios improve their near-term realism.

## 2   PIAM validation tool

### 2.1   Overview

The package *piamValidation* provides validation tools for the Potsdam Integrated Assessment Modeling environment. Developed in R, this open-source package is freely available on GitHub under the LGPL-3 license[1], facilitating transparency and collaboration in research and analysis.

Users provide the tool with IAM scenario data and relevant reference data for historical or future time periods. Criteria for checks are defined in a configuration file to perform a vetting. This configuration file serves as a flexible interface, which allows for various use cases:

- – Deviations of scenario to reference data

- – Deviations between:

    - – Models (same scenario, same periods);

    - – Scenarios (same model, same periods);

    - – Periods (same model, same scenario).

- – Conformity to scalar thresholds, such as an absolute value or growth rate.

Deviations can be defined as either absolute or relative deviations. The function `validateScenarios` takes the inputs mentioned above and evaluates the validation checks. In the standard setting, this results in a dataset that contains an evaluation of either "green" (test passed), "yellow" (warning), or "red" (fail) for each data point to be evaluated. To further enhance visualization, additional colors can be employed to distinguish between exceedance of lower bounds and upper bounds. In that

---

[1]https://github.com/pik-piam/piamValidation

case, "blue" and "cyan" are used to hint toward values below thresholds, and "yellow" and "red" remain to capture exceeded upper thresholds.

The function `validationHeatmap` makes these results accessible by arranging them for one variable on an interactive heat map using the R library *ggplotly*. Depending on the category of checks performed, the underlying data used in the evaluation can be viewed for each data point by hovering over the corresponding square in the heat map. Finally, `validationReport` can be used to perform the validation calculation and plot all heat maps in one HTML report, which is automatically rendered from an RMarkdown file provided by the package.

After successfully installing the *piamValidation* R package in an appropriate R environment, the tool structure can be described along three dimensions: Input, data processing, and output, as illustrated in Figure 1.

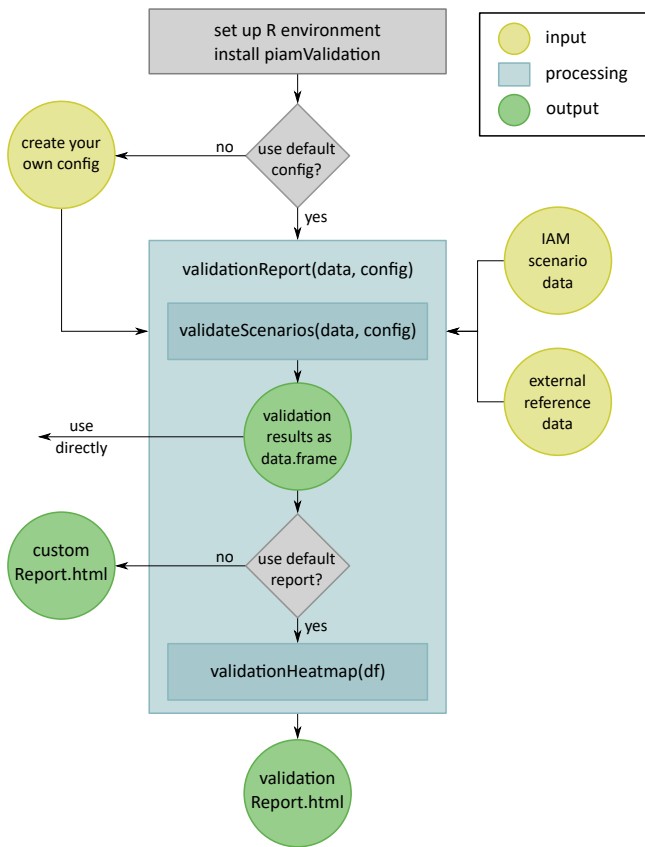

**Figure 1.** Validation workflow with respective inputs, processing steps and outputs.

## 2.2  Data Input

The input data for the *piamValidation* tool comprises three key components. First, the IAM scenario data to be validated. Second, the reference data, which is used to compare the initial IAM scenario data. This reference data may include outputs

from other IAM models or scenarios, as well as third-party external datasets, such as historical, observational or other prediction data. Third, a configuration file is necessary to specify the details of the validation. The *piamValidation* R package provides a standard set of configuration files for general use. However, users seeking greater flexibility can create and customize their own configuration files to address specific validation requirements.

To facilitate use by various IAM communities and external users, IAM data are organized according to the standardized Integrated Assessment Consortium (IAMC) format[2]. It organizes data along the dimensions of "model", "scenario", "variable", "region" and "period". Thus, the validation process works with 5-dimensional data objects, expecting a data point to be uniquely defined as $x(m, s, v, r, p)$, with the indices referring to these five dimensions.

### 2.2.1 Configuration File

The *piamValidation* framework allows for a wide range of different validation approaches. They are defined via a configuration file with predefined columns as described in Table 1. This is intended to enable users with limited programming expertise to use the tool, as the structure of the validation configuration file follows the intuitive framework of a validation process, answering the following three guiding questions:

1. Which type of validation should be performed using which metric?

2. Which data should be validated?

3. What is the reference being used for the validation?

These questions are answered by filling in the columns of the configuration file as described in Table 1.

---

[2]https://www.iamconsortium.org/scientific-working-groups/data-protocols-and-management/iamc-time-series-data-template/

| Column | Description |
|---|---|
| **metric** | Decide which type of comparison to perform, currently supported: "relative", "difference", "absolute", "growthrate". |
| **critical** | Is it considered to be a critical check? The function `validationPass` will report failed checks only if this is set to "yes". |
| **variable** | Choose one or multiple variables to be checked: Define multiple variables via "*" (one sub-level) or "**" (all sub-levels), e.g. "Final Energy\|*" for "Final Energy\|Industry" and "Final Energy\|Transportation", etc. and "Final Energy\|**" for "Final Energy\|Industry\|Electricity", "Final Energy\|Industry\|Electricity\|Cement" and so on. |
| **unit** | This is an optional column that allows to verify that scenario - and reference data are consistent in their use of units. |
| **model** | Choose one or multiple (comma-separated) models, or leave empty to choose all. |
| **scenario** | Choose one or multiple (comma-separated) scenarios, or leave empty to choose all. |
| **region** | Choose one or multiple (comma-separated) regions, or leave empty to choose all. |
| **period** | Choose one or multiple (comma-separated) periods, or leave empty to choose all (generally "all" defaults to =< 2100 in this case, for historical checks 2005-2020). A range of periods can be defined via "yyyy-yyyy". |
| **min/max_yel/red** | Define minimum and maximum thresholds which decide whether a check is passed (green), produces a warning (yellow) or is failed (red). If reference data is missing, the result will be gray. Each line needs at least one threshold. Relative thresholds can be given either as percentage (e.g., 20%) or decimal (e.g., 0.2). |
| **ref_model** | For model intercomparison, set one or multiple reference models here, all models chosen in the column "model" will be compared to it. |
| **ref_scenario** | This column can be used in two ways - either compare two scenarios produced by the same model to one-another (similar to the model intercomparison), or set it to *historical* to compare model data to observational data provided by one or multiple external sources. It is recommended to choose a historical reference source explicitly in the "ref_model" column for historical comparisons. |
| **ref_period** | Compare data between different periods, set one or multiple reference periods here. |
| **notes** | An optional column that allows leaving comments about the thresholds or checks in this row. |

**Table 1.** Column description of configuration file. Note: Each described column must be filled in accordance to the use case to ensure successful operation of the *piamValidation* tool.

### 2.2.2 Validation types and metrics

A common validation procedure consists of comparing IAM results to benchmarks[3], such as observational data from external data sources. Such reference data can be handed to the validation tool in large quantities by supplying a CSV, Excel, or MIF file that is structured along the same five dimensions as the IAM data, with the specific requirement that the "scenario" column reads "historical" for all data points. This way, the tool can also differentiate between a model intercomparison exercise, where the same scenario is compared between two or more models, and a comparison of multiple scenarios to one or multiple reference models. In terms of data dimensions, a relative deviation can thus be expressed as

$$\text{rel\_deviation} = \frac{x(m_i, s_j, v_k, r_l, p_m) - x(m_{\text{ref}}, s_{\text{hist}}, v_k, r_l, p_m)}{x(m_{\text{ref}}, s_{\text{hist}}, v_k, r_l, p_m)} \ ,$$

where $x(m_i, s_j, v_k, r_l, p_m)$ are the IAM values using the notation introduced in 2.2 and $x(m_{\text{ref}}, s_{\text{hist}}, v_k, r_l, p_m)$ are the observational data values from one or multiple reference models. In an analogous way, a "difference to historical" check can be performed by defining an absolute deviation to the reference data as

$$\text{abs\_deviation} = x(m_i, s_j, v_k, r_l, p_m) - x(m_{ref}, s_{hist}, v_k, r_l, p_m).$$

Similarly, checks with the metric "relative" or "difference" can also be performed against subsets of the scenario data itself, which then functions as reference data. This can in turn be performed in three different ways - comparing to a selected period, scenario, or model:

$$\text{abs\_deviation\_period} = x(m_i, s_j, v_k, r_l, p_m) - x(m_i, s_j, v_k, r_l, p_{ref}) \ ,$$
$$\text{abs\_deviation\_scen} = x(m_i, s_j, v_k, r_l, p_m) - x(m_i, s_{ref}, v_k, r_l, p_m) \ or$$
$$\text{abs\_deviation\_model} = x(m_i, s_j, v_k, r_l, p_m) - x(m_{ref}, s_j, v_k, r_l, p_m).$$

Relative comparisons again divide this difference by the reference value. Only one dimension in period, scenario, or model can be used as a validation dimension, while the other two must remain constant. Accordingly, it is not possible to compare scenario $s_j$ of model $m_i$ with scenario $s_{ref}$ of model $m_{ref}$). The reference of this dimension can be either a single period/scenario/model or a comma-separated list of them. Furthermore, when choosing multiple sources the user can decide whether the validation thresholds should be based on the *range* of the reference values, where the maximum value of the group is used as the reference for upper and the minimum as the reference for lower thresholds or their *mean* will serve as reference for all thresholds.

For instance, in a near-term validation study, multiple data sources can be chosen as references by selecting the following expression in the config file as ref_model[4]:

$$range(m_i, m_j, ..., m_x).$$

---

[3]The practice of benchmarking models, periods, or scenarios against one specific element of the respective dimension is referred to as intercomparison throughout the remainder of this text.

[4]Note: The same holds for ref_scenario and ref_period.

The metric "absolute" does not require any reference data, as it contains the absolute values of the thresholds directly in the threshold columns of the configuration file.

Lastly, the "growthrate" metric can be used to perform validation checks with respect to average yearly growth rates. For a time step of $dt$ between periods, the growth rate is calculated as

$$150 \quad \text{growthrate} = \left( \frac{x(m_i, s_j, v_k, r_l, p_m)}{x(m_i, s_j, v_k, r_l, p_{m-dt})} \right)^{\frac{1}{dt}} - 1 \;\; .$$

Depending on the metric of the validation check and whether it is a comparison to observational data, different columns of the configuration file need to be filled as shown in Figure 2.

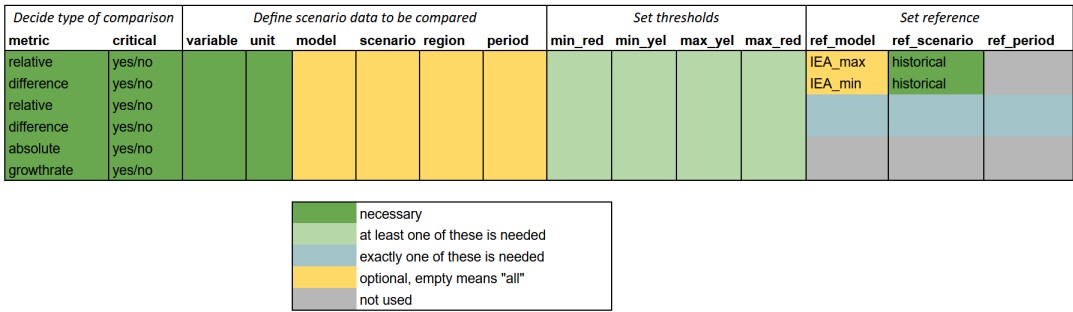

**Figure 2.** Overview of required and optional columns of the configuration file, depending on metric.

## 2.3   Data Processing

The validation configuration file allows users to define a wide range of validation checks, going beyond the ones provided
by default in the validation tool. On the one hand, this requires the data processing and evaluation to be both consistent and flexible, e.g. when performing data harmonization as n intermediate calculation steps might be required in some validation exercises. On the other hand, users might supply the validation tool with data and configuration files that are either internally inconsistent or exceed the scope of the currently supported use cases. In such cases, it is crucial to provide transparency by ensuring the user can identify the issue through clear mechanisms and the inclusion of essential information in error messages.
These requirements are met by the modular structure of the tool, where each data processing step is carefully separated from the next. The structure of the central data processing function of the validation tool, `validateScenarios`, is sketched using pseudocode in Algorithm 1.

     At the beginning of a validation process using the `validateScenarios` function, the configuration file and scenario and reference data are imported and checked for consistency (see Algorithm 1). The loop starting in line 4 takes information from
one row of the configuration file and assembles the scenario and reference data needed for the validation check defined. It connects the thresholds that are relevant for each data point to the scenario data. In case a validation check is performed against a reference value (either historical or period/scenario/model intercomparison), this value is appended for each data point. The

resulting data slices for all rows are combined and checked for any duplicates that can result from overlapping definitions in the configuration file. Lines 11 and 12 contain the calculation and evaluation of the actual validation check, according to the respective metrics. Finally, data can be either exported as CSV or returned as an R data.frame.

---

**Algorithm 1** validateScenarios is the core of the piamValidation framework.

---

**Input:** scenarioData, referenceData, validationConfig

**Output:** data.frame or CSV file containing original data, reference values and validation results

**function** `validateScenarios()`

1: data = read(scenarioData, referenceData)

2: cfg = read(validationConfig)

3: cfg = check, clean and expand cfg

4: **for all** row in cfg **do**

5:     slice data according to filters in row

6:     left join config thresholds to data slice

7:     left join reference values to data slice (if applicable)

8:     valiData = bind all data slices

9: **end for**

10: valiData = removeDuplicateRows(valiData)

11: calculate value to be checked against threshold (depending on metric)

12: evaluate check and append validation result (color) to valiData

13: (optional: export valiData as CSV file)

14: return(valiData)

---

## 2.4 Output and Visualization

While the data output of `validateScenarios` can be used in any preferred way, two specific visualization approaches are offered to users of *piamValidation*. The function `validationHeatmap` provides a quick overview of the general outcome of a validation by laying out the validation result colors across up to four dimensions. This means that the five-dimensional IAM data objects need to be reduced to a single element in only one dimension. By default, the "variable" dimension is chosen for this. An example can be seen in Figure 3 where "region" and "period" are chosen for the x- and y-axis respectively and "scenario" and "model" are plotted as x- and y-facets.

Alternatively, the function `linePlotsThresholds` offers an easy way to plot the thresholds as colored background areas behind line plots of the scenario data and, if applicable, the corresponding reference data. For this type of plot, one additional data dimension needs to be singular, by default, this falls to the "region" dimension.

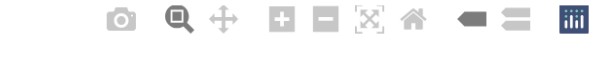

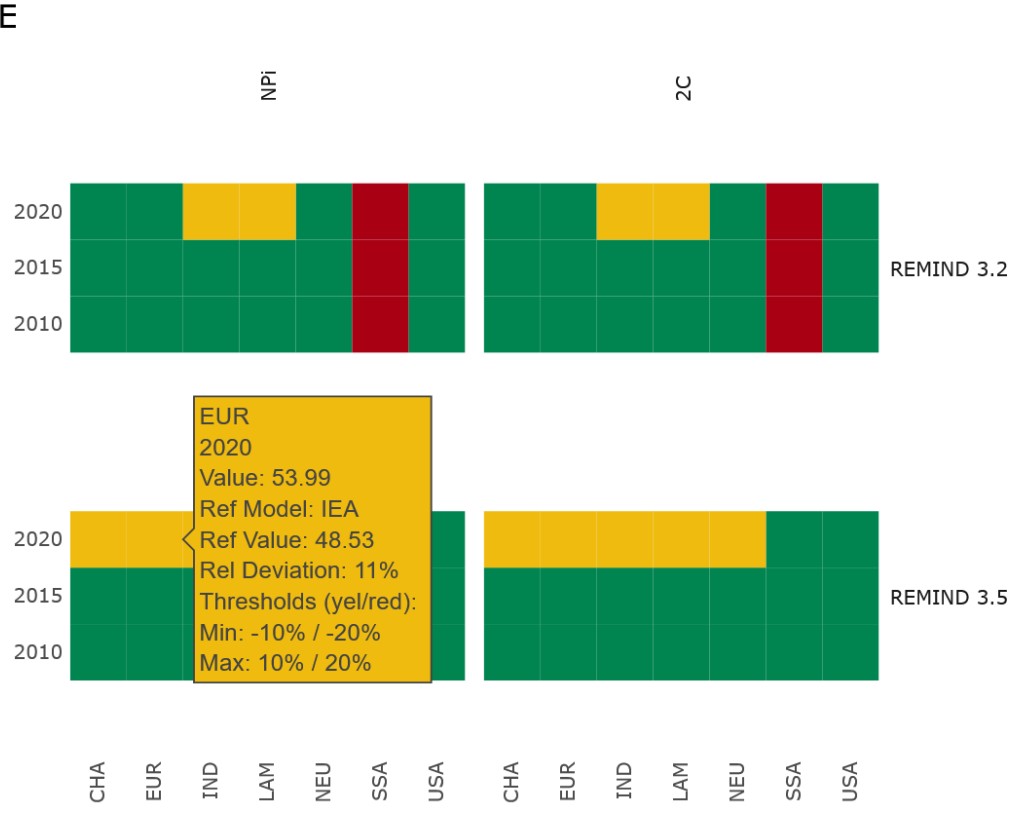

**Figure 3.** Example of an interactive validation heat map. Data along four dimensions. Hovering over a data point shows additional information in a tooltip. In this example, the Final Energy (FE) in two versions of the REMIND model for two different policy scenarios (National Policies implemented (NPi) and "well below" 2 degrees (66% likelihood) of temperature rise by the end of the century (2C) are compared with the historical IEA data.

The heat map plotting routine makes use of the *ggplot2* library using the `geom_tile` function in combination with *plotly* which enables interactive access to additional information by showing tooltips when hovering over a tile. This allows users to first get a quick overview of the general validation outcome by scanning the colors of the heat map and identifying areas of interest (e.g., a region with many red tiles). In a second step, the exact value of a data point as well as the corresponding thresholds and reference values, including their origin, are shown in the tooltip. This two-step process is a particular strength of the chosen visualization approach, as it makes validation results of large datasets accessible without overcrowding the plot or distributing the data over many different plots (line plots are usually not useful for visualizing data along four dimensions, often it is the "region" dimension which is spread over multiple plots).

The heat map tiles are chosen to always be squares, which results in the plot layout being dependent on the lengths of the data dimensions. In cases where a high number of scenarios or models is being validated, this can lead to gaps between heat

map facets and inefficient use of space in the validation document. To alleviate this, `validationHeatmap` allows the user to manually specify the arrangement of the data dimensions by defining the function arguments "x_plot", "y_plot", "x_facet" and "y_facet". If not set manually, the function determines the optimal layout by analyzing the lengths of the data dimensions.

## 2.5 Accessibility

One of the primary objectives of the *piamValidation* package is to enable broad application of the tool and support IAM capacity building for developers and users of scenarios. To achieve this, the tool is published under an open-source license on GitHub and designed with user-friendliness as a central design criterion. For basic usage, a validation process can be executed with a single command, ensuring accessibility and ease of use for a broad range of users.

For the first application, users can follow four straightforward steps: First, ensure a working R environment. If it is not already installed, one possibility could be to install R and the integrated development environment RStudio [5].

Second, install the *piamValidation* package by running the following R command:

```
install.packages("piamValidation", repo = "https://rse.pik-potsdam.de/r/packages")
```

The third step consists of the data preparation. Verify that the IAM data that needs to be validated, as well as the reference data, follow the IAMC standard format. In addition, prepare the configuration file by defining the validation checks to be performed as described in Section 2.2.1 and Section 2.2.2.

In the final step, the following R command must be executed to generate the HTML validation report, including interactive heat maps. This command consolidates the file paths for the IAM data to be validated, the reference data, and the appropriate configuration file into a single operation.

```
piamValidation::validationReport(c("path_to_IAM_data","path_to_ref_data"), "path_to_config")
```

For advanced users, the post-processing and visualization can be modified according to individual validation requirements by using the `R data.frame` containing the validation results directly. In addition, we encourage users to participate in ongoing discussions and package developments on GitHub:

https://github.com/pik-piam/piamValidation/issues.

## 3  Application case

We examine two application examples. The first, based on NGFS scenarios, emphasizing how *piamValidation* can be applied to an existing set of scenarios from different models, illustrating the range of potential analyses. The second demonstrates how *piamValidation* can be applied to assess and enhance the near-term realism of technology trends in the REMIND model. To support these applications, *piamValidation* offers two visualization options. We initially employ heat maps to obtain a

---

[5]Download the freeware here: R https://www.r-project.org/ and RStudio https://posit.co/products/open-source/rstudio/

preliminary indication of deviations. Complementarily, line plots offer a more granular perspective by depicting the detailed behavior of individual variables and their temporal dynamics.

## 3.1 NGFS scenarios

To demonstrate the methodological versatility of the *piamValidation* tool, we perform a series of validation exercises using the NGFS scenarios[6] (Richters et al., 2024). This choice is motivated by two central considerations. First, these scenarios are not only widely recognized within the integrated assessment modeling and academic communities but are also extensively used by policymakers, financial supervisors, and other stakeholders to assess climate-related risks and transition pathways. Second, the scenarios are developed with multiple IAMs (MESSAGE[7], GCAM[8] and REMIND[9]), making them particularly suitable for illustrating the ability of *piamValidation* to address heterogeneous modeling frameworks.

This validation exercise is a qualitative one, focusing on the type of checks performed rather than the exact selection of threshold values. This implies that threshold exceedance does not indicate limitations in the scenario data but rather illustrate how the tool can be used to identify specific patterns. Furthermore, the plots use additional colors to indicate whether upper or lower thresholds are crossed via the function argument `extraColors = TRUE` when calling `validateScenarios`. The corresponding validation configuration file for these application cases and the markdown file to create the plots below are available via GitHub[10].

### 3.1.1 Validation overview for multiple models

As a primary application type, the *piamValidation* tool can be used to validate multiple variables of different models against historical reference data and near-term estimates.

The heat map overview (Figure 4a) allows a quick identification of areas of concern. In this case, one conclusion that can be drawn from this visualization is that the near-term dynamics of wind capacities are consistently flagged across all models: while GCAM tends to overestimate the 2025 data point across all scenarios, MESSAGE and REMIND seem to rather underestimate the expansion of wind until 2030, especially in scenarios with weaker climate ambition. Using the *piamValidation* plotting function for time series `lineplotThreshold`, Figure 4b provides a more detailed assessment of the variable.

---

[6]Data available here: https://zenodo.org/records/13989530

[7]MESSAGEix-GLOBIOM: Model for Energy Supply Strategy Alternatives and their General Environmental Impact - GLObal BIOsphere Model.

[8]GCAM: Global Change Analysis Model.

[9]REMIND-MAgPIE: REgional Model of INvestments and Development - The Model of Agricultural Production and its Impact on the Environment.

[10]https://github.com/pik-piam/piamValidation/blob/main/inst/config/validationConfig_publication_NGFS.csv

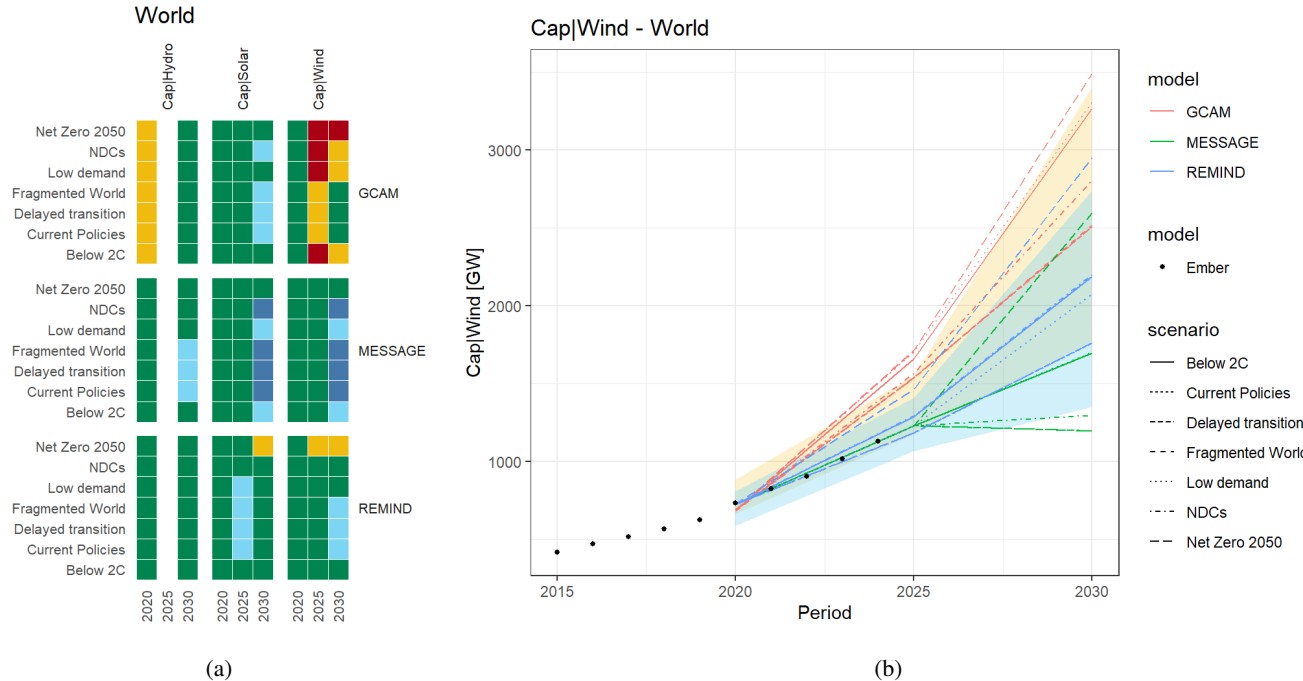

**Figure 4.** Example of a validation exercise that starts by looking at a) data of multiple variables, models, scenarios, and periods via validation heat map while reducing the region dimension to one element ("World"). After identifying a variable of interest, b) takes a closer look at wind capacities by performing a line plot of scenario data alongside historical reference data and the validation thresholds as colored funnels.

### 3.1.2 Model intercomparison

The piamValidation package also allows for model intercomparison exercises by selecting one model as the reference (here: MESSAGE). In this example, we examine global $CO_2$ emissions and identify occurrences of REMIND or GCAM deviating more than 20% (weak threshold) or 40% (strong threshold) from MESSAGE within each scenario. The heat map in Figure 5a reveals that the strongest deviations appear after 2050, with REMIND and GCAM showing lower emissions than MESSAGE. However, as emissions drop closer to zero, the relative differences being used as thresholds make up smaller absolute values. This becomes clearer when looking at a line plot of a specific scenario (here: "Delayed transition") and seeing a "closing" funnel (see Figure 5b).

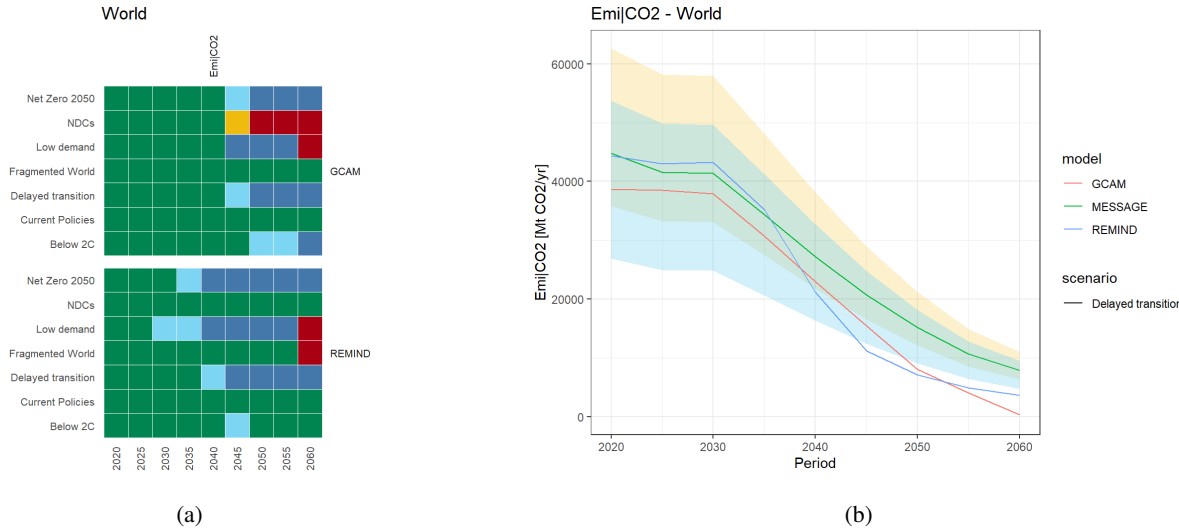

(a) (b)

**Figure 5.** Model intercomparison example showing a) a heat map of the NGFS data set looking at relative deviations of $CO_2$ emissions from REMIND and GCAM compared to MESSAGE next to b) the line plot for the Delayed Transition scenario with a funnel of $\pm$ 20% (green area) and 40% (yellow/cyan area) around the MESSAGE data.

Users who want to avoid this effect can choose the "difference" metric instead of the "relative" one to define constant thresholds around the reference model. Applying a buffer of $\pm$ 5/10 $GtCO_2$/yr results in a validation outcome as shown in Figure 6a and 6b.

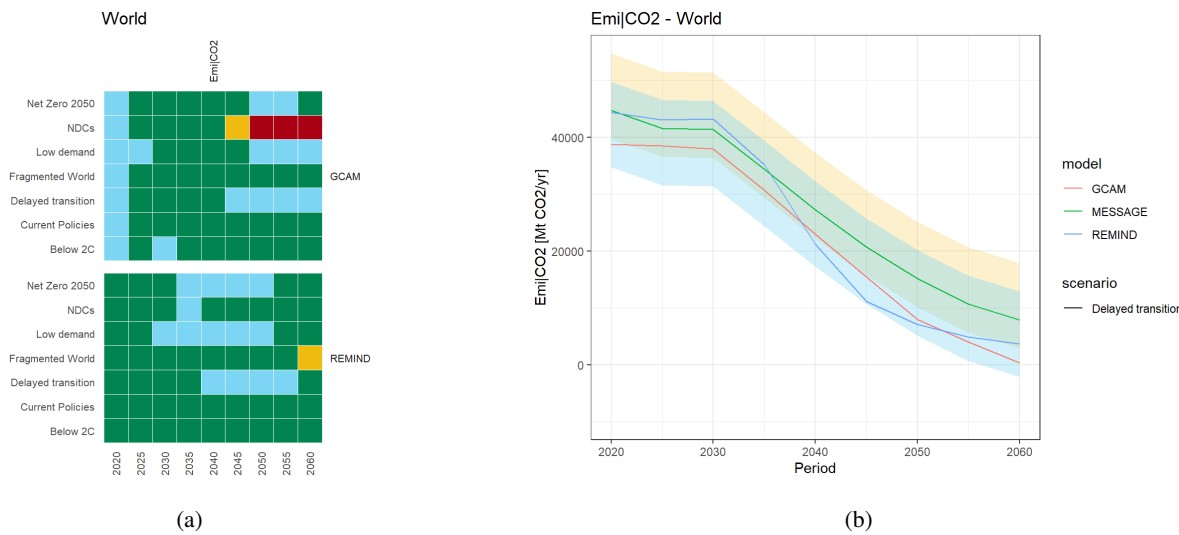

(a) (b)

**Figure 6.** Model intercomparison example now showing a) a heat map of the NGFS data set looking at constant deviations to MESSAGE $CO_2$ emissions and b) the line plot for the Delayed Transition scenario with a funnel of fixed width of $\pm$ 5 $GtCO_2$/yr (green area) and $\pm$ 10 $GtCO_2$/yr (yellow/cyan area) around the MESSAGE data.

### 3.1.3  Scenario intercomparison

Similarly, scenarios can also be compared with each other (here: the reference is the "Below 2C" scenario). Consistent with the underlying scenario narratives, more ambitious scenarios such as "Net Zero 2050" and "Low Demand" are characterized by lower $CO_2$ emissions, whereas less ambitious scenarios such as "Current Policies" and "Fragmented World" exhibit higher $CO_2$ emissions. This application case as shown in Figure 7a and 7b can serve as a straightforward means of conducting a preliminary plausibility check of scenario narratives.

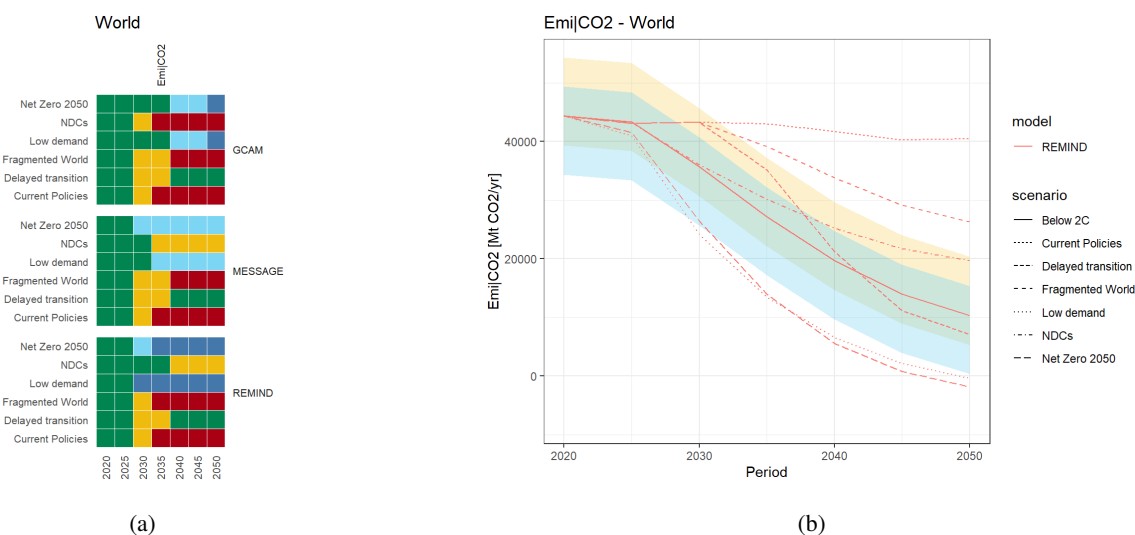

(a) (b)

**Figure 7.** Scenario intercomparison example with a) a heat map showing which scenarios are higher or lower in emissions compared to the "Below 2C" scenario for each model and b) a line plot for REMIND only, with constant width funnels of $\pm$ 5/10 GtCO$_2$/yr around the reference scenario.

### 3.1.4  Period intercomparison

Finally, periods can also be evaluated in relation to one another. This example checks how emissions of the periods 2025, 2030, and 2035 compare to the 2020 period. This approach can be used to validate that emissions in ambitious climate policy scenarios should decline within a plausible reduction rate. In Figure 8a and 8b upper thresholds are 0% and +10% for all validated periods, while lower thresholds go from -15% to -30% (weaker thresholds) and -30% to -60% (stronger thresholds) from 2025 to 2035. As all scenarios start at the same 2020 emissions value, their funnels are also identical, which allows them to be plotted together. Note that this case also demonstrates the option of choosing asymmetrical thresholds. The results indicate that in the net zero and the low demand scenario, emission reductions are considerably more pronounced in the REMIND model compared to GCAM or MESSAGE.

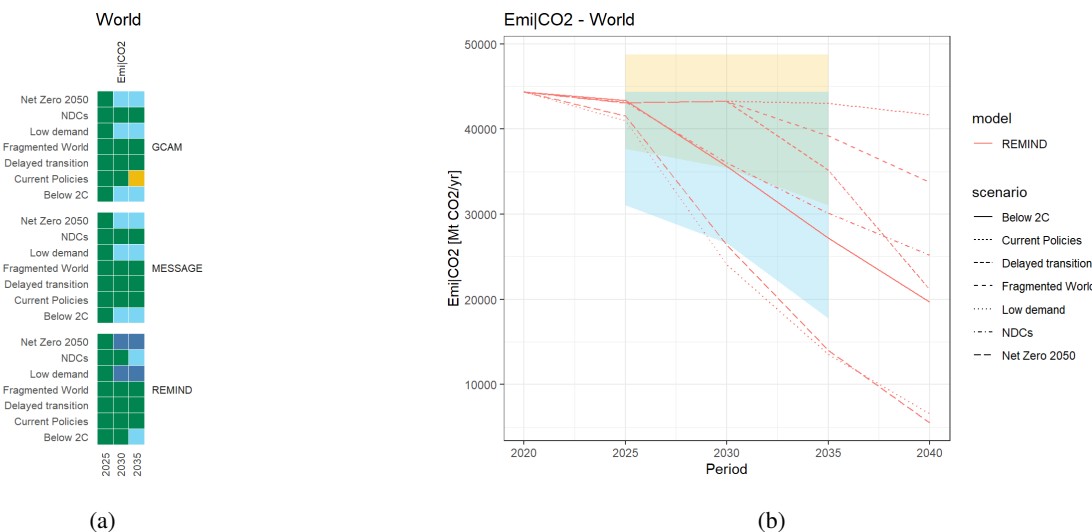

**Figure 8.** Period intercomparison example demonstrating a feasibility study on how fast global emissions can be reduced compared to 2020, using a) a heat map for all models and b) a line plot for REMIND.

## 3.2 REMIND Validate short-term technology trends

This section presents an application of the *piamValidation* tool to the Integrated Assessment Model REMIND (see Baumstark et al. (2021) for a detailed description). To illustrate the validation process, we validate model scenarios against historical data and observational or third-party technology trend data until 2030, focusing on three key technology trends: capacity of $CO_2$ transport and storage (CTS), sales and stock of battery electric vehicle (BEV), and capacity of offshore wind power. These technology trajectories represent some of the key energy system transformations that Luderer et al. (2022) describe: the potential for carbon management, the electrification of end-use sectors, and the composition of the electricity mix.

In scenarios of the REMIND 3.2.0 release (Luderer et al., 2023), all three technologies exhibit significant deviations compared to historical and reference data. Upon identifying significant deviations, the focus shifts to improving the model and refining its underlying assumptions. The key strategies for achieving these improvements can be summarized as follows:

1. Updating the model input data;

2. Validating socio-economic assumptions to ensure alignment with observed trends;

3. Revising and verifying associated cost estimates;

4. Revising the representation of the inertia in up-scaling new technologies.

All adjustments have been incorporated into the REMIND 3.5.0 release (Luderer et al., 2025). We select two climate policy scenarios spanning a wide range of policy ambition: National Policies implemented (NPi) and "well below" 2 degrees (66%

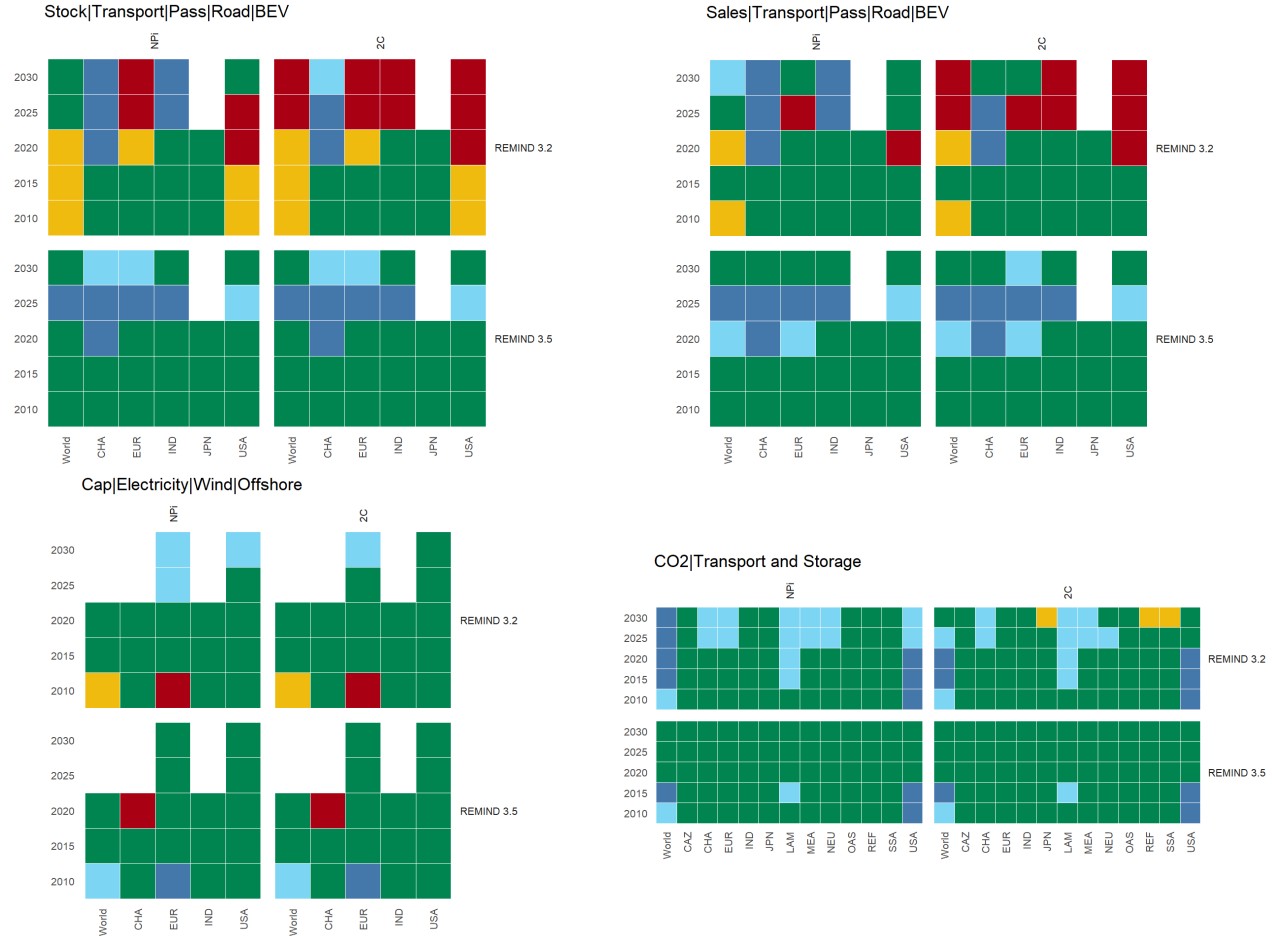

**Figure 9.** *piamValidation* **heat maps of selected technology trends.** Each heat map presents the performance of REMIND 3.2.0 (top) compared to the updated version, REMIND 3.5.0 (bottom) for the two climate policy scenarios: National Policies implemented (NPi, left) and "well below" 2 degrees (2C, right). For BEV we validate historical modeling values against IEA Global Electric Vehicle Outlook (GEVO) historical data. For the outlook, the modeling values are validated against the IEA GEVO Announced Pledges Scenario (APS) and Stated Policies Scenario (STEPS). Similarly, wind offshore capacity is historically compared to International Renewable Energy Agency (IRENA) data, and the outlook validated Global Wind Energy Council (GWEC) data. Carbon management capacity is validated against the IEA CCUS Projects Database. An overview of all thresholds is provided in the validation configuration file (https://github.com/pik-piam/piamValidation). In addition, a detailed description and community discussion on the thresholds can be accessed here: https://github.com/pik-piam/mrremind/discussions/544. In general, regional bounds are set more generously than global bounds, reflecting higher uncertainties and potential variations in accounting methods

likelihood) of temperature rise by the end of the century (2C)[11]. Both scenarios follow the Shared Socioeconomic Pathways 2 (SSP2) called Middle of the Road. The 2°C stylized climate policy scenario assumes a peak budget of 1150 $GtCO_2$ and 1000 $GtCO_2$, respectively on total $CO_2$ emissions from 2020 to 2100. Figure 9 illustrates how the model enhancements improve the near-term realism measured as performance against the reference data within the *piamValidation* tool.

More dynamic insights on model to trend deviation are presented in Figure 10. The key developments in the updated model are described below.

### 3.2.1 $CO_2$ transport and storage

$CO_2$ transport and storage comprise the capture, transport, and storage or use of $CO_2$ from different $CO_2$ sources. All $CO_2$ that is captured, transported, and stored is accounted for in the variable *"CO2|Transport and Storage"*. The carbon transport and storage infrastructure is represented as a single technology in REMIND. The construction of geologic carbon storage infrastructure is associated with long lead times, as the development of geologic storage typically takes 7 to 10 years from exploration to industrial-scale injection (Bui et al., 2018). Upscaling may be further limited by the availability of geological engineers and drilling rigs (Budinis et al., 2018). In addition, a wide range of cost estimates have been documented. While studies on technical cost range between 3 and 20 USD/$tCO_2$ (Budinis et al., 2018; Ipcc, 2014), costs announced for projects reach up to 80 USD/$tCO_2$ (Bellona, 2022; Jakobsen et al., 2017; IOGP, 2023).

There are thus two trends relevant for model improvement. First, storage potentials until 2030 are limited by projects announced today due to the long lead times. We thus refer to the database on Carbon Capture, Utilization, and Storage (CCUS) projects, published by the International Energy Agency (IEA) in 2023 that is also used for the validation reference values. To improve the near-term realism of geologic storage in the model, we introduce regional lower and upper bounds. The lower bounds in 2025 and 2030 are based on the announced capacities of storage projects that are either operational or under construction (with announced starts of operation before 2025 and 2030, respectively). The upper bounds for 2025 and 2030 additionally include 40% of the capacity of all announced projects that report a start of operation by 2025 or 2030, respectively. Note that for these, the final investment decision is often pending, thus a certain share of unrealized project capacities is assumed based on historic delay and failure rates. Furthermore, as the EU27 pursues a transnational, Europe-wide CCS infrastructure, the capacity of a given CCS project is not fully attributed to the respective member state within the model. Instead, the EU27 CCS capacities are pooled (including 50% of Norway's and 50% of UKs near-term potential) and redistributed across member states based on GDP. The second improvement concerns the costs of carbon transport and storage. The technical cost estimate in REMIND 3.2.0 of 7.5 USD/$tCO_2$ appears too low, despite REMIND modeling the cost of upscaling through adjustment cost. We thus increase our capital cost estimate to reach 12 USD/$tCO_2$ informed by the average in Budinis et al. (2018).

---

[11]The exact original scenario names are "SSP2EU-NPi" and "SSP2EU-PkBudg1150" in REMIND 3.2.0 and "SSP2-NPi2025" and "SSP2-PkBudg1000" in REMIND 3.5.0 for NPi and 2C respectively. They have been renamed for clarity and brevity.

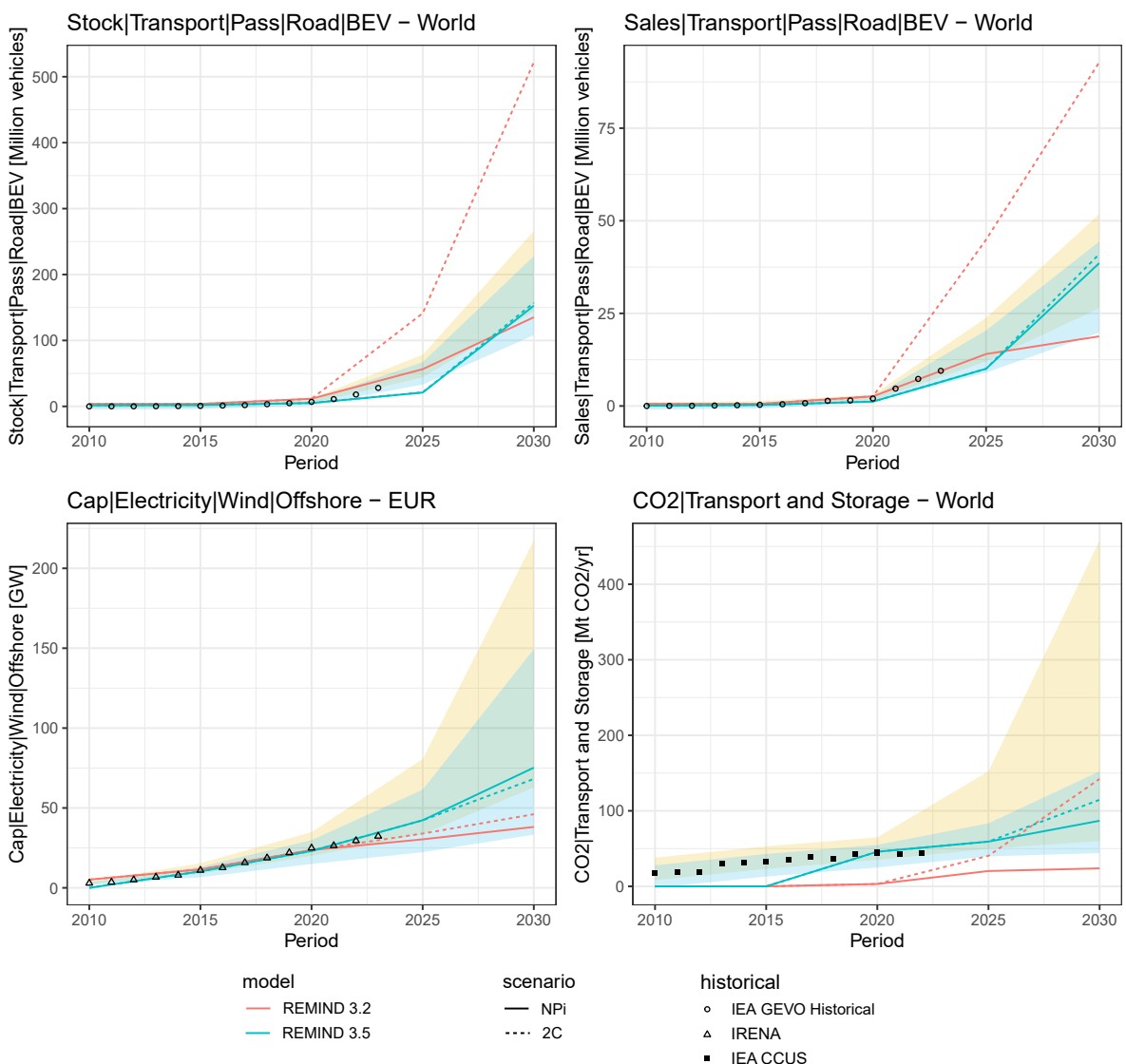

**Figure 10.** Historical and near-term realism. The REMIND scenarios and thresholds presented are identical to those in Figure 9. Scenarios updated refer to the improvements in REMIND 3.5.0

### 3.2.2 Electric vehicle stock and sales

REMIND's transportation sector is modeled by the simulation model "Energy Demand Generator-Transport" (EDGE-T) (Rottoli et al., 2021) that is integrated into the broader sociotechnical framework of REMIND. EDGE-T simulates technology and transport mode decisions relying on a multinomial logit approach. The decision tree, the logit, comprises a nested structure of subsequent choices among comparable alternatives for transport modes and technologies. This established approach is suited for the modeling of end-consumer decisions under probabilistic conditions and allows for including non-economic factors, such as inconvenience costs, in the decision simulation.

In the transport sector, a prominent example is the increase in electric vehicles (EVs), which is not mainly driven by $CO_2$ prices and, consequently, fuel costs, but rather by technical innovations leading to up-front investment cost reductions, policy incentives, and shifts in how consumers assess this technology alternative. The piamValidation tool revealed deviations from near-term estimates for stocks and sales of EVs. Model parameters were thus updated to reflect technological advancement and better understanding of the uptake dynamics of electric vehicles. Building on the Global EV Outlook (IEA, 2024) and its information about the shares of electric vehicles in car sales and car fleets in different world regions, the model representation of consumer attitudes (so-called inconvenience costs) towards electromobility was updated. Furthermore, input data processing was critically assessed, e.g., by screening for implausible outliers in input data, and correction routines were implemented. As high-quality transport data, such as energy service demand, vehicle fleet characteristics, and vehicle costs, are scarce, verifying data plausibility across countries, technologies, and vehicle sizes is crucial for improving model and scenario quality.

### 3.2.3 Offshore wind capacity

The competition between onshore and offshore wind power is largely determined by factors like grid integration costs or the matching between generation and demand. REMIND 3.2.0 – like other long-term global integrated assessment models – represents these drivers in a very aggregated and parameterized way, since it does not explicitly resolve the relevant spatial and temporal scales. Consequently, the model limits the share of offshore wind generation in total wind energy to a certain range. This share is constrained in the near-term to reflect the historical lag between onshore and offshore deployment.

In REMIND 3.2.0, historical capacities are available until 2020, at which point offshore wind is still a nascent technology with around 35 GW of installed capacity worldwide. To prevent the model from deploying offshore earlier than observed in the real-world, only a portion of the total offshore wind potential is available: before 2010, none of the potential is available, then the available portion increases gradually until 2050. Conversely, specific equations ensure that the development of offshore capacity is not too slow compared to onshore after 2025: in a given region, if the available offshore potential is four times as high as onshore, the equations constrain the model to deploy each year at least twice as much offshore as onshore new capacity.

The *piamValidation* tool has revealed that these constraints were overly pessimistic: wind offshore deployment in REMIND 3.2.0 is slower than observed in real data, particularly in Europe. To address this discrepancy, we updated the model to match the real data more closely. REMIND 3.5.0 uses newly released data by IRENA (2024) to compute consolidated historical capacities for 2020 and a lower bound for 2025 based on 2023 capacity. Moreover, the model allows for an accelerated phase-

350 in of offshore wind by making the full potential available by 2030 instead of 2050. The equations linking offshore to onshore are maintained after 2030 in order to follow the rising trend of offshore wind despite its higher investment costs.

## 4 Conclusions

Integrated Assessment Models are central to the climate negotiations and play a key role in the IPCC assessments and thus at the interface between climate science and policy. However, they face challenges and criticism regarding their transparency
(Robertson, 2021), their ability to capture technology trends (Keppo et al., 2021; Anderson, 2019) and their overall reliability (Wilson et al., 2021).

We introduce the open-source R package *piamValidation*, a versatile tool designed to address multiple use cases in the validation of IAM scenario data. This package facilitates validation across IAM scenarios, models, and temporal periods. Furthermore, it supports validation against historical, observational, and benchmark datasets, using time series, thresholds, or
360 growth rates as a reference. To enhance reliability, *piamValidation* can be seamlessly integrated into the regular development workflow of IAMs, enabling comprehensive sanity checks and performance evaluations.

We outline the structure of the *piamValidation* package, emphasizing its user-friendly design to facilitate widespread adoption within the IAM community and further stakeholders. The aim is to enhance the overall validity and reliability of IAMs through the systematic application of the tool. For this purpose, we provide a structured guideline together with standard use
cases, demonstrating the application of the tool through single-command execution.

We demonstrate the versatility of the *piamValidation* tool in two complementary application cases. They highlight how the tool can be used broadly across contexts as well as to refine the near-term dynamics of technology pathways in REMIND. The REMIND case study demonstrates that the usefulness of the tool for historical and benchmark validation critically depends on the chosen validation settings and the availability of reliable reference data. Consequently, the utility of the *piamValidation* tool
would be enhanced by broader adoption and the systematic documentation of configuration files within the IAM community.

The primary benefit of the *piamValidation* tool lies in its ability to efficiently provide an overview of variable deviations within large-scale datasets. It also facilitates systematic validation across IAM scenarios and supports sanity checks throughout model development processes. Furthermore, the tool serves as a potential platform for the IAM community to collaboratively build a repository of knowledge on reasonable validation thresholds.

Despite its utility, the *piamValidation* tool has some important limitations that fall into three broad areas. First, the quality of any validation exercise cannot be guaranteed by the tool itself but instead depends on user and community choices. Most importantly, the identification of meaningful validation cases, the selection of appropriate reference data, and the definition of reasonable thresholds all substantially influence the outcome of the validation exercise. Although these challenges are not technical limitations of the piamValidation tool, they substantially influence the quality, consistency, and acceptance of the
validation results. Second, certain limitations arise from the design of the tool. In particular, caution is required when thresholds are defined in terms of relative deviations and the validation values approach zero. Under such conditions, even minimal absolute differences can manifest as disproportionately large relative deviations, complicating the interpretation of results. In

these cases, the use of absolute deviations is preferable. Finally, the tool is constrained by challenges in data management. Harmonizing scenario data with reference sources often requires substantial effort to ensure consistency in units, definitions,
sectoral coverage, and technological detail. This integration process can be time-consuming.

Ongoing improvements are being made to the *piamValidation* tool, with continuous development efforts focused on enhancing transparency of error handling, unit testing of various use cases and additional capabilities. These include three central development areas: First, we combine relative and absolute metrics to address scale dependency in relative deviations, particularly near zero. To reduce this scale dependency and increase the user-friendliness of the *piamValidation* tool, we are working
on a new feature that includes a general tolerance. We plan to implement an optional lump-sum buffer, defined as a fraction of the largest reference value, within the processing workflow. Second, we plan to enable differentiation across validation criteria, such as feasibility, sustainability, or equity. A third key development area centers around the ability to handle, compare and vet scenarios from an extensive scenario database. These improvements are designed to make the tool more user-friendly and versatile for the IAM community.

In addition to these development areas, we anticipate a process of continuous improvement for the validation of realism in technology trends. Central to this is the ongoing expansion of reference variables and the refinement of existing thresholds, which requires regularly updating current data sources while monitoring and integrating new ones. Further developments may include extending validation metrics to assess mid-term feasibility and policy robustness. The extent and pace of these improvements will be accelerated by the tool's widespread application, particularly through the sharing of configuration files
and active participation in GitHub discussions, which facilitate feedback, highlight priorities, and guide future enhancements.

*Code and data availability*. The REMIND-MAgPIE code is implemented in GAMS, whereas code and data management is done using R. The REMIND 3.2.0 code is archived at Zenodo (https://zenodo.org/records/7852740) Luderer et al. (2023) and REMIND 3.5.0 at (https://zenodo.org/records/15147820) Luderer et al. (2025). The technical model documentation is available on the common integrated assessment model documentation website (https://www.iamcdocumentation.eu/Model_Documentation_-_REMIND-MAgPIE), and is pub-
lished as open-source code (https://github.com/remindmodel/remind. under the GNU Affero General Public License v3.0).

A repository for the source code of the *piamValidation* package is available via GitHub at https://github.com/pik-piam/piamValidation under the GNU Lesser General Public License v3.0. The *piamValidation* code is archived at Zenodo under (https://zenodo.org/records/17661999) Weigmann et al. (2025).

The reproduction of the plots in this publication is documented at https://pik-piam.github.io/piamValidation/articles/publication.html
and can be achieved with the RMarkdown files at https://github.com/pik-piam/piamValidation/blob/main/inst/markdown/validationReport_publication.Rmd and https://github.com/pik-piam/piamValidation/blob/main/inst/markdown/validationReport_publication_NGFS.Rmd. The validation in this study is performed with the configuration files at https://github.com/pik-piam/piamValidation/blob/main/inst/config/validationConfig_publication.csv and https://github.com/pik-piam/piamValidation/blob/main/inst/config/validationConfig_publication_NGFS.csv.

*Author contributions.* GL, LB and EK supervised the development of the *piamValidation* package. The *piamValidation* package is initiated and led by PW, where FL and OR contributed to its development. FL, AM, TD, JH, JM, RP, OR, LV, NB, FB, CC, RM and GL contributed to the REMIND model development. All authors commented, reviewed and contributed to the final manuscript.

*Competing interests.* One of the authors, Gunnar Luderer, serves as a Topic Editor for this journal. All other authors declare no competing interests.

*Acknowledgements.* The authors are grateful to the attendees of the Seventeenth Assessment Modeling Consortium (IAMC) Annual Meeting 2024 for their valuable feedback and reflections. This work received support from the European Commission Directorate-General of Climate Action (DG CLIMA) under Service Contract No. 14020241/2022/884157/SER/CLIMA.A.2 CLIMA/2022/EA-RP/0007 and the Horizon Europe Project PRISMA, under the grant agreement number 1010816.

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
