# Peer review of "Validation of climate mitigation pathways"

_EGUsphere, 2025_

## Author Comment (AC1)

Response to Anonymous Referee #1

*Throughout the document, the original comments of the anonymous referee are presented sequentially in **black** and italic font.*

The authors' responses are provided in **blue** font to ensure clear distinction.

***Comment:** This reviewer's understanding of piamValidation and the study presented in 'Validation of climate mitigation pathways' is that the package serves as a crucial tool for improving the reliability of Integrated Assessment Models (IAMs). While IAMs are essential for shaping climate policy, they often face criticism for their lack of transparency and their limited ability to account for real-world technological advancements. piamValidation aims to address these concerns by systematically comparing IAM scenario data against historical observations, feasibility limits, and other model results. The package is designed for ease of use, requiring minimal coding to generate interactive HTML reports with heat maps, which encourages broader adoption and the development of more realistic near-term scenarios. Its effectiveness is demonstrated using the REMIND model, highlighting its ability to detect emerging technological trends that diverge from expected patterns—such as developments in carbon dioxide transport and storage, electric vehicles, and offshore wind power. Clear visual feedback, including 'traffic light' evaluations, helps model developers implement meaningful improvements.*

**1 Response:** We would like to thank reviewer #1 for the comprehensive and insightful review. The level of involvement and depth of feedback has substantially helped us enhance the quality of our manuscript.

***Comment:** If this interpretation is correct, then this reviewer has identified weaknesses and several areas for improvement.*

*First, the scope of validation variables and case studies appears limited. The current application primarily focuses on select technologies and the REMIND model. To enhance the tool's applicability and robustness, this reviewer suggests expanding its scope to include other sectors, variables and case studies with different IAMs (e.g. MESSAGE, GCAM; https://www.ngfs.net/ngfs-scenarios-portal/glossary/#IAM). A broader range of applications would help demonstrate the versatility of piamValidation and strengthen its reliability across diverse modeling frameworks.*

**1.1 Response:** Thank you for this valuable comment. We agree that additional application cases can further demonstrate the tool's applicability and strengthen its robustness. Therefore, in the revised manuscript, we will add a section on the general applicability of the tool. In addition, we support the reviewer's suggestion to apply the tool to open-source NGFS scenarios, and we show an example below.

**NGFS Application**

The following paragraph demonstrates the versatility of piamValidation by performing four types of validation checks on the NGFS scenarios v5.0 (https://zenodo.org/records/13989530). This validation exercise is a qualitative one, focusing on the type of checks performed rather than the exact selection of threshold values. This implies that threshold violations do not indicate limitations in the scenario data but rather illustrate how the tool can be used to identify specific patterns. Furthermore, the plots use additional colors to indicate whether upper or lower thresholds are violated via the function argument "*extraColors = TRUE*" when calling "*validateScenarios*".

The corresponding validation configuration file for these application cases and the markdown file to create the plots below are available on GitHub piamValidation.

1. Validation overview for multiple models.

The heat maps of the validation report are able to represent four dimensions: in Figure 1 below for instance, the dimensions are scenarios, years, variables and models. The model's dimension is shown when another dimension has only one value, either the region (here: "World") or the variable. One conclusion that can be drawn from this visualization is that the near-term dynamics of CCS are consistently flagged across all models: most tend to underestimate the 2020 data point while overestimating the 2030 value in many scenarios.

[Figure]

Figure 1: NGFS model overview

**2. Model intercomparison**

The piamValidation package allows for model intercomparison exercises by selecting one model as the reference (here: MESSAGEix-GLOBIOM). In this example, we examine global CO2 emissions and identify occurrences of REMIND or GCAM deviating more than 20% (weak threshold) or 40% (strong threshold) from MESSAGE within each scenario.

The heat map in Figure 2 a) reveals that the strongest deviations appear after 2050, with REMIND and GCAM showing lower emissions than MESSAGE. However, as emissions drop closer to zero, the relative differences being used as thresholds make up smaller absolute values. This becomes clearer when looking at a line plot of a specific scenario (here: "Delayed transition") and seeing a "closing" funnel (see Figure 2 b).

a)                                        b)

[Figure]

*Figure 2 NGFS relative model intercomparison in relation to MESSAGEix-GLOBIOM*

Users who want to avoid this case can choose the "difference" metric instead of the "relative" one to define constant thresholds around the reference model. Applying a buffer of +/- 5/10 Gt CO2/yr results in a validation outcome as shown in Figure 3 a) and b).

[Figure]

*Figure 3 NGFS absolute model intercomparison in relation to MESSAGEix-GLOBIOM*

**3. Scenario intercomparison**

In a similar fashion, scenarios can also be compared with each other (here: the reference is the "Below 2C" scenario). Consistent with the underlying scenario narratives, more ambitious scenarios such as "Net Zero 2050" and "Low Demand" are characterized by lower $CO_2$ emissions, whereas less ambitious scenarios such as "Current Policies" and "Fragmented World" exhibit higher $CO_2$ emissions. This application case can serve as a straightforward means of conducting a preliminary plausibility check of scenario narratives.

[Figure]

*Figure 4 NGFS scenario intercomparisopn*

**4. Period intercomparison**

Finally, periods can also be selected in relation to one another. This example checks whether the periods 2025 and 2030 compared to the period 2020 are between -20% and 0% (weak thresholds) or -40% and +10% (strong thresholds). Note that this case also demonstrates the option of choosing asymmetrical thresholds.

a)                                         b)

[Figure]

*Figure 5 NGSF period intercomparison*

*Comment: Second, the effectiveness of the validation is inherently dependent on data quality and methodological transparency. The reliability of the process hinges on the accuracy and robustness of observational and benchmark data. This reviewer calls for a more detailed discussion on managing uncertainties in reference datasets, along with a deeper technical explanation of how validation thresholds are determined, particularly for complex or uncertain data. In addition, this reviewer suggests that the authors incorporate metadata quality indicators for input reference datasets and establish a shared, moderated repository of standard validation thresholds to enhance transparency and reproducibility.*

**1.2 Response:** Thank you for this valuable remark. In the revised version of the manuscript, we will place greater emphasis on how the usefulness of the tool and the quality of its results depend on the reference data employed and the thresholds selected.

Through the tool's configuration file, the applied thresholds and data source references are published and ensure reproducibility. An open discussion with other IAM teams and interested stakeholders regarding the selection of data sources and the determination of thresholds has already been initiated for the REMIND application case, as documented here: https://github.com/pik-piam/mrremind/discussions.

To tackle the uncertainty around specific sources, the tool is able to take several sources for the same threshold. With the configuration ref_model "***range(sourceA, sourceB)***", the flag will appear only when the data is beyond the threshold for sourceA *and* for sourceB. Use-cases include allowing different models to rely on different sources or smoothing yearly data so that they ignore short-term disruptions (like the 2020 variations due to the pandemic).

In addition, the platform https://pik-piam.github.io/piamValidation/ provides systematic procedures and standardized benchmarks to ensure transparency and reproducibility of the results.

*Comment: Third, technical barriers and user accessibility warrant further consideration. While piamValidation is designed for ease of use, its reliance on R and the IAMC data format may present challenge\*s for users unfamiliar with these tools. To broaden accessibility, this reviewer suggests providing more guidance for non-R users or exploring interfaces for alternative platforms, such as Python. Expanding compatibility across multiple programming environments would help ensure that a wider audience - including researchers and policymakers with varying technical backgrounds - can effectively use the tool.*

**1.3 Response:** Yes, we agree that the tool should also be used for non-R users. As the purpose of the piamValidation tool is to validate IAM data, the validated data should be in the standard IAM format, as this is the format of open IAM scenario data. The manuscript can be used as a step-by-step tutorial for non- R users as it explains the steps for:

1. Installing R environment: install R and the integrated development environment RStudio 4. Download the freeware here: R https://www.r-project.org/ and RStudio https://posit.co/products/open-source/rstudio/

2. Installing piamValidation:
   *install.packages("piamValidation",repo="https://rse.pik−potsdam.de/r/packages")*
3. Single execution command:
   *validationReport(c("path_to_IAM_data","path_to_ref_data"),"path_to_your_config")*

This procedure enables users with no prior R experience to install the necessary environment, run *piamValidation*, and obtain a full validation report with minimal interaction.

**Comment:** *Fourth, the future directions are not entirely clear. While ongoing development is mentioned, a more structured roadmap outlining planned enhancements would be beneficial. This reviewer suggests specifying future improvements, such as incorporating machine learning techniques, expanding the range of validation variables and integrating the tool with additional modelling frameworks. In addition, extending validation metrics to assess long-term feasibility and policy robustness would strengthen the tool's relevance for decision-making in climate policy.*

**1.4 Response:** We appreciate the reviewer's thoughtful comment and will restructure the outlook in the revised manuscript. In terms of future developments, the highest priority is the expansion of validation variables and refinement of existing thresholds. This requires regularly updating the data sources that we currently use and monitoring and evaluating new data sources to integrate them into the tool.

Rather than focusing on expanding the validation tool towards other languages or data frameworks, the development direction is set on making the tool more stable, precise, and intuitive to use. At this stage, we do not see a clear benefit from incorporating machine learning methods and prefer to rely on established data sources. Although highly desirable, extending the validation period to enable a long-term feasibility assessment in this application is currently constrained by the limited availability of reliable reference data.

First attempts were started to compare IAM scenarios to other energy-focused projections, such as the IEA "Net Zero by 2050" scenario. However, due to methodological differences, e.g., in sector definitions, these comparisons have sparked limited interest so far.

**Comment:** *Fifth, the discussion on tool limitations lacks depth. A more comprehensive examination of potential biases, uncertainties and challenges in applying piamValidation across diverse IAMs would strengthen the manuscript. This reviewer suggests providing clearer guidelines for identifying and mitigating these issues, ensuring that users can navigate the tool's constraints effectively. Addressing these limitations in greater detail would enhance transparency and reinforce the reliability of validation outcomes.*

**1.5 Response:** Thank you for pointing this out. We restructure the section on limitation by providing a clearer distinction into the following three aspects:

1. Limitations indirectly related to the piamValidation tool,

2. tools limitation,
3. limitations regarding data management.

This restructuring will be similar to:

The piamValidation tool is subject to several limitations that fall into three broad areas. First, many aspects are indirectly related to the tool itself and instead depend on user and community choices. For instance, the identification of meaningful validation cases, the selection of appropriate reference data, and the definition of reasonable thresholds all substantially influence the outcome of the validation exercise. Although these challenges are not technical limitations of the piamValidation tool, they substantially influence the quality, consistency, and acceptance of the validation results. Second, certain limitations arise from the design of the tool. In particular, caution is required when thresholds are defined in terms of relative deviations and the validation values approach zero. Under such conditions, even very small absolute deviations can manifest as disproportionately large relative differences, complicating the interpretation of results. In these cases, the use of absolute deviations is preferable. Finally, the tool is constrained by challenges in data management. Harmonizing scenario data with reference sources often requires substantial effort to ensure consistency in units, definitions, sectoral coverage, and technological detail. This integration process can be time-consuming.

*Comment: Finally, additional specific comments are provided in the annotated manuscript file.*

**1.6 Response:** Thank you for the detailed language review. The revised manuscript will be reformulated accordingly where appropriate.

*Comment: This reviewer offers an overall endorsement and recommends acceptance, contingent on minor revisions to address the areas for improvement outlined above. These revisions would further enhance the paper's contribution to strengthening the credibility of IAMs.*

---

## Author Comment (AC2)

Response to Anonymous Referee #2

*Throughout the document, the original comments of the anonymous referee are presented sequentially in **black** and italic font.*

The authors' responses are provided in **blue** font to ensure clear distinction.

***Comment****: In order to limit temperature rise, it is essential to explore pathways for reducing greenhouse gas emissions. The methodology presented in this paper is considered valuable in contributing to the transparency of scenarios derived from integrated assessment models (IAMs) that estimate future GHG emission pathways. Therefore, the paper is deemed worthy of publication.*

**2 Response:** We are grateful to Reviewer #2 for the detailed assessment and thoughtful feedback, which have been instrumental in enhancing the quality of our paper. We provide responses to the individual comments under their corresponding paragraphs below.

*However, the following points should be addressed prior to publication.*

- ***Comment****: The development of integrated assessment models (IAMs) to support greenhouse gas mitigation strategies in developing countries is expected to become increasingly important.*

  *International databases such as those provided by the IEA are widely referenced in the development of IAM scenarios in developed countries. However, a key challenge going forward is how researchers in developing countries can secure access to such existing data, and how they can collect and make use of country-specific information on technologies and socio-economic conditions.*

  *Therefore, if the piamValidation proves effective in enabling researchers in developing countries to conduct practical scenario development, it would be of considerable significance. In this regard, it would be highly valuable for readers if the paper could elaborate on how such existing data can be utilized by researchers in developing countries.*

**2.1 Response:** We appreciate this important comment. A central goal of piamValidation is to lower the barrier for researchers and stakeholders worldwide in validating IAM scenario data. In this context, the configuration files include collections of open-source reference data, which simplify both data search and preparation. This approach significantly reduces the time and resource requirements for conducting such analyses.

- ***Comment****: The paper lacks sufficient explanation as to why CCS (Carbon Capture and Storage) is considered a viable option in short- and medium-term scenarios.*

  *In both the United States and Australia, several CCS projects have failed. According to*

*a report by the U.S. Government Accountability Office (GAO), seven out of eight CCS projects supported by the Department of Energy (DOE) were canceled. The only project that became operational, Petra Nova, saw NRG Energy withdraw from the project, with JX Nippon acquiring full ownership and shifting to a sole operation structure. Although the facility was restarted in September 2023, its commercial viability remains uncertain. CCS alone does not generate profit. Moreover, if the oil produced through Enhanced Oil Recovery (EOR) is combusted, additional $CO_2$ is emitted, raising questions—particularly from a life cycle assessment (LCA) perspective—about whether CCS via EOR leads to a net reduction in greenhouse gas emissions. The IEA's Net Zero by 2050 scenario calls for the storage of 1.5 billion tonnes of $CO_2$ per year by 2030. However, the IEA itself acknowledges that achieving this would require "unprecedented investment and policy support" and that current progress is "seriously off track." This paper also references the long lead time associated with CCS deployment.*

*Given these points, analyzing CCS as a short- to medium-term mitigation option seems questionable. If CCS is to be proposed as a viable strategy within this timeframe, the paper should include a detailed explanation of where and under what conditions it could realistically be implemented.*

**2.2 Response:** We thank the reviewer for raising this critical point: Historic upscaling of CCS has been much slower than expected in IAM scenarios that follow the stringency of the climate targets set by the Paris Agreement. This is, as the reviewer pointed out, in large part due to the lack of real-world policy support, as CCS does not generate profit by itself, alongside other challenges in implementation. Although the last years have seen substantial new activities and investments (Northern Lights, 2024; Porthos, 2023; Reuters, 2022), near-term deployment will be limited. This makes it all the more important to include this variable in piamValidation, as IAMs have in the past substantially overestimated the potential for massive near-term upscaling of CCS (Fuhrman et al., 2025; Kazlou et al., 2023; Zhang et al., 2024). PiamValidation will flag such overoptimistic upscaling by comparing scenario results against the CCUS database and thus may help modelers implement more realistic upscaling dynamics.

- ***Comment****: While the adoption of electric vehicles (EVs) is progressing in certain regions, there are still areas where uptake remains limited due to persistent consumer concerns such as high costs, range anxiety, and constraints in battery supply. How does the analysis in the paper account for these regional differences? Furthermore, for regions where challenges remain, can the authors provide proposals or suggestions for addressing these issues?*

**2.3 Response:** *Thank you for this important remark. Besides vehicle investment and operational cost, it is crucial to include local preferences or inconvenience costs shaped, e.g., by infrastructure availability, model availability, and risk aversion towards new technologies for modeling the adoption of electric vehicles in different regions. These assumptions were regionally updated with the help of piamValidation and the comparison to the scenarios*

*presented in the global EV outlook (IEA, 2024). For a detailed description of the modeling of electric vehicle adoption in EDGE-Transport, the authors refer to Rottoli et al., (2021).*

- ***Comment****: Offshore wind power has also faced negative public perceptions, including concerns about low-frequency noise. More recently, rising material and labor costs have led to higher bid prices and an increasing number of project cancellations. How does piamValidation address or account for these challenges?*

**2.4 Response:** The validation bounds for wind development are based on external data sources: IRENA (International Renewable Energy Agency) for the current capacity GWEC (Global Wind Energy Council), and BNEF (Bloomberg New Energy Finance) for upcoming additions until 2030. For a more detailed overview on the calculation of the boundary thresholds, please see https://github.com/pik-piam/mrremind/discussions. The validation tool allows for small deviations around current capacities because different models may use sources with slightly different accounting methods. As future development is inherently uncertain, the validation tool adds margins around the projections of external sources: the lower bound, for instance, assumes a high failure rate of announced projects. At the regional level, we acknowledge that the uncertainty is even higher, partly indeed due to labor costs and public perception, and partly due to different region definitions in models and data sources; the validation bounds are therefore much looser and only trigger yellow flags and no red flags.

- ***Comment****: To effectively reduce GHG emissions, demand-side mitigation plays a significant role. Although the paper focuses primarily on technologies, when comparing scenarios, it is also important to consider how demand-side reductions are treated. Please also address demand-side mitigation.*

**2.5 Response:** We thank the reviewer for also considering the demand-side mitigation pathways. The piamValidation tool can also be used to validate near-term demand trends, under the condition that reliable external data is available to assess future demand. Such data is unfortunately limited, and it remains unclear whether observed changes in demand occur autonomously or result from policy interventions or social dynamics. By contrast, electric vehicles and heat pumps represent trillion-dollar markets, which has led international organizations and corporations such as the IEA and BNEF to invest considerable resources in producing and updating detailed market forecasts. These forecasts in turn provide benchmarks for piamValidation to evaluate the short-term realism of IAM pathways.

Previous studies on demand-side mitigation strategies (Muessel et al., 2025; van Heerden et al., 2025) find that electrification is the main driver of greenhouse gas reduction, which is why integrated assessment modeling focuses on improving its representation. However, additional research on demand-side mitigation is needed to understand its effects on emissions and other indicators (Creutzig et al., 2022), as well as to provide reliable reference data.

- *Comment*: *The paper states that "IAM scenarios contribute to the IPCC assessments," but the primary purpose of IAMs is to support policy formation, not to contribute directly to the IPCC.*

  *Indeed, IAM analyses are an important component of IPCC reports, and being reviewed and synthesized by the IPCC helps convey findings widely to policymakers, which is highly meaningful.*

  *However, it is important to clarify that the original role of IAM scenario development is to present options and impact assessments for real-world policy challenges, maintaining its independent purpose.*

**2.7 Response:** We agree with this assessment and will amend the misleading wording in the revised manuscript.

- *Comment:* *"The explanation of how scenarios can be improved through the use of the piamValidation is insufficient."*

**2.8 Response:** Thank you for this remark. In the revised manuscript, we will further improve the description of the piamValidation applicability, focusing on two aspects:

1. **Demonstrating versatility through additional application cases**: We add a section that illustrates the versatility of piamValidation by applying it to open-source NGFS scenarios. This section demonstrates how the tool can be employed for multi-model overviews, and for the intercomparison of models, scenarios, or periods. Further details are provided in the response letter to Referee 1, Response 1.1.

2. **Clarifying limitations and user responsibility**: Furthermore, we want to stress the limits of the piamValidation tool and precisely point out how and where the responsibility of the user starts in a refined description of the tool. Further details are provided in the response letter to Referee 1, Response 1.5.

- *Comment:* *The statement "Early IAM applications date back to the late 1990s (Cointe et al., 2019)" appears to be inaccurate; it would be more appropriate to say "the late 1980s" or "the early 1990s."*

  *For instance, James Edmonds and his colleague published Global Energy: Assessing the Future (Oxford University Press, New York) in 1985, in which they analyzed future pathways using a model.*

  *In addition, the IPCC First Assessment Report (AR1) published in 1990, the Supplementary Report published in 1992, and Climate Change 1994: Radiative Forcing of Climate Change and An Evaluation of the IPCC 92 Emissions Scenarios all*

*included analyses of future GHG emissions pathways and their implications for temperature projections.*

*Furthermore, regarding the IMAGE model referenced by Cointe et al. (2019), IMAGE 1.0 was developed by Rotmans in 1990, and IMAGE 2.0 was edited by Alcamo in 1994.*

**2.9 Response:** Thank you for pointing this out and providing the detailed explanation. We entirely agree and will correct this in the revised manuscript.

**References**

Creutzig, F., Niamir, L., Bai, X., Callaghan, M., Cullen, J., Díaz-José, J., Figueroa, M., Grubler, A., Lamb, W.F., Leip, A., Masanet, E., Mata, É., Mattauch, L., Minx, J.C., Mirasgedis, S., Mulugetta, Y., Nugroho, S.B., Pathak, M., Perkins, P., Roy, J., de la Rue du Can, S., Saheb, Y., Some, S., Steg, L., Steinberger, J., Ürge-Vorsatz, D., 2022. Demand-side solutions to climate change mitigation consistent with high levels of well-being. Nat. Clim. Chang. 12, 36–46. https://doi.org/10.1038/s41558-021-01219-y

Fuhrman, J., Lane, J., McJeon, H., Iyer, G.C., Edwards, M.R., Thomas, Z., Edmonds, J.A., 2025. Rate and growth limits for carbon capture and storage. Environ. Res. Lett. 20, 064034. https://doi.org/10.1088/1748-9326/add9af

IEA, 2024. Global EV Outlook 2024 – Analysis.

Kazlou, T., Cherp, A., Jewell, J., 2023. Feasible deployment trajectories of carbon capture and storage compared to the requirements of climate targets. https://doi.org/10.21203/rs.3.rs-3275673/v1

Muessel, J., Pietzcker, R., Hoppe, J., Verpoort, P., Klein, D., Luderer, G., 2025. An integrated modeling perspective on climate change mitigation and co-benefits in the transport sector. Environ. Res. Lett. 20, 094011. https://doi.org/10.1088/1748-9326/adf23f

Northern Lights, 2024. Accelerating decarbonisation. URL https://norlights.com/what-we-do/

Porthos, 2023. CO2 reduction through storage under the North Sea. URL https://www.porthosco2.nl/en/project/

Reuters, 2022. China's CNOOC completes first offshore carbon capture site. URL https://www.reuters.com/business/environment/chinas-cnooc-completes-first-offshore-carbon-capture-site-2022-06-15/

Rottoli, M., Dirnaichner, A., Pietzcker, R., Schreyer, F., Luderer, G., 2021. Alternative electrification pathways for light-duty vehicles in the European transport sector. Transportation Research Part D: Transport and Environment 99, 103005. https://doi.org/10.1016/j.trd.2021.103005

van Heerden, R., Edelenbosch, O.Y., Daioglou, V., Le Gallic, T., Baptista, L.B., Di Bella, A., Colelli, F.P., Emmerling, J., Fragkos, P., Hasse, R., Hoppe, J., Kishimoto, P., Leblanc, F., Lefèvre, J., Luderer, G., Marangoni, G., Mastrucci, A., Pettifor, H., Pietzcker, R., Rochedo, P., van Ruijven, B., Schaeffer, R., Wilson, C., Yeh, S., Zisarou, E., van Vuuren, D., 2025. Demand-side strategies enable rapid and deep cuts in buildings and transport emissions to 2050. Nat Energy 1–15. https://doi.org/10.1038/s41560-025-01703-1

Zhang, Y., Jackson, C., Krevor, S., 2024. The feasibility of reaching gigatonne scale CO2 storage by mid-century. Nat Commun 15, 6913. https://doi.org/10.1038/s41467-024-51226-8

---

## Author Comment (AC3)

Response to Anonymous Referee #3

*Throughout the document, the original comments of the anonymous referee are presented sequentially in **black** and italic font.*

The authors' responses are provided in **blue** font to ensure clear distinction.

***Comment****: The manuscript presents the development and application of piamValidation, an open-source R package aimed at enhancing the transparency and credibility of integrated assessment models (IAMs). This tool enables structured comparisons of scenario data against historical trends, feasibility bounds, and across models, thereby addressing well-known criticisms related to transparency and technological realism. The application to the REMIND model demonstrates its practical relevance and potential to strengthen confidence in IAM-based analyses. Given the importance of IAMs in shaping climate policy, systematic validation tools are highly valuable.*

**3. Response:** We thank the anonymous reviewer for the assessment of our manuscript and for the constructive recommendations on how it can be further improved.

*The manuscript would benefit from addressing the following points to strengthen its suitability for publication:*

*1)The current implementation of piamValidation is applied primarily to a limited set of technologies within the REMIND model. This narrow focus restricts the demonstration of the tool's broader applicability. To enhance generalizability and robustness, the validation scope could be expanded to include a wider array of sectors, variables, and IAMs;*

**3.1 Response**: We thank the reviewer for highlighting this limitation. We agree that the presentation of the piamValidation tool would benefit from additional application cases illustrating its use with other models and contexts. Accordingly, we will include an additional section in the revised manuscript applying the tool to open-source NGFS scenarios. This section demonstrates how the tool can be employed for multi-model overviews, and for the intercomparison of models, scenarios, or periods. Further details are provided in the response letter to Referee 1, Response 1.1.

*2)The discussion of the tool's limitations is somewhat superficial. A deeper analysis of possible uncertainties, and the difficulties of applying piamValidation to various IAMs would enhance the strength of the manuscript.*

**3.2 Response**: We thank the reviewer for pointing this out. We restructure the section on limitation in the revised manuscript by providing a clearer distinction into the following three aspects:

1. Limitations indirectly related to the piamValidation tool,
2. tools limitation,
3. limitations regarding data management.

Thereby we seek to clearly highlight the limits of the tool's functionality and indicate where user responsibility begins. Further details are provided in the response letter to Referee 1, Response 1.5.

---

## Author Response (AR1)

**Relevant changes in the manuscript**

The following changes were performed in the manuscript to address the reviewers comments:

1. The largest addition to the updated manuscript consists of an additional application case of the validation tool, applying it to the NGFS scenarios as proposed by the reviewers. We use the new chapter 3.1 to shortly present the NGFS project before outlining multiple use cases which are covered by existing features of piamValidation. The newly included use cases are:

   - Multi-model, multi-variable overview
   - Model intercomparison
   - Scenario intercomparison
   - Period intercomparison

   They are presented alongside 5 new figures, each consisting of a heat map for a general overview and a line plot demonstrating a focused analysis. The abstract, introduction and conclusions were adjusted accordingly to reflect the inclusion of the additional application case.

   Furthermore, we chose to activate the *extraColors* feature for all heat maps in chapter 3, which uses blue and cyan to signal the exceedance of lower bounds, allowing a clearer distinction of which thresholds were surpassed.

   With this additional application case we incorporate the following reviewer comments into the revised manuscript:

   Reviewer 1: Response 1.1, Reviewer 2: Response 2.8, Reviewer 3: Response 3.1 and 3.2

2. We changed the structure in outlining the limitations of the tool by separating between the following areas:
   - Limitations indirectly related to the piamValidation tool,
   - tool limitations,
   - limitations regarding data management.

   With this improvement we incorporate the following reviewers' comments into the revised manuscript:

   Reviewer 1: Response 1.5, Reviewer 2: Response 2.8, Reviewer 3: Response 3.2

3. In addition to the limitations, we reworked the future developments and put special focus on the expansion and curation of reference data and thresholds.

   This is in response to:

   Reviewer 1: Response 1.2 and 1.4

4. To simplify the reproduction of the plots, a new vignette has been added to the R package and deployed to Github pages. The link to the vignette was added to the Code availability statement together with links to the newly created data and validation configuration files used in the new application case.

   https://pik-piam.github.io/piamValidation/articles/publication.html

   With this improvement we incorporate the following reviewers' comments into the revised manuscript:

   Reviewer 1: Response 1.3

5. Overall, we clarified the language and explanations, harmonized formatting and fixed minor typos. Furthermore, we specified which REMIND versions were used exactly in application case 2.

   With this improvement we incorporate the following reviewers' comments into the revised manuscript:

   Reviewer 1: Response 1.6, Reviewer 2: Response 2.7, 2.8 and 2.9

---

## Referee Report (RR1)

**Referee report for "Validation of climate mitigation pathways"**

The manuscript introduces and details the *piamValidation* package, developed to "enable systematic comparison of variables" from IAM datasets. The package is presented as an instrument that can improve reliability and transparency of IAMs, and two uses are highlighted: the comparison and harmonization of scenarios across IAMs, and the comparison of near-term dynamics in IAM scenarios with external, historical datasets. Two applications, corresponding to these two use cases, are then presented: one on NGFS scenarios, highlighting the different types of comparisons/harmonization that the tool can perform, and one for the improvement of the REMIND model through the validation of near-term technology dynamics.

The manuscript is clearly written and provides a good overview, description and discussion of the package. As someone who has some knowledge of the IAM landscape but is not a modeller, I found it easy to follow and understand, which suggests it is suitable for the introduction of a package that is presented as user friendly.

Below are a few points that deserve clarification and/or discussion.

- Section 3.2 "REMIND Validate short-term technology trends" presents an application of the *piamValidation* package to REMIND scenario data and discusses subesquent changes between REMiND 3.2.0 and REMIND 3.5.0 to improve the accuracy of the near-term technological dynamics in the model, but it is not always clear how exactly the use of *piamValidation* informed the changes. This is especially the case in subsections 3.3 and 3.4 where the outcomes from the use of *piamValidation* are not mentioned explicitly, so it does not come out clearly how the changes implemented relate to the diagnosis (as opposed to updating/refining in the light of recent literature or data only). Section 3.5 is much clearer in that respect.
- Very minor point on this section, but for consistency the subsections should be labelled "3.2.1/3.2.2/3.2.3" rather than "3.3/3.4/3.5".
- The introduction refers to "the key evaluation criterion outlined by Wilson et al. (2021)". It would need to briefly state what these criteria are and briefly explain how the *piamValidation* tool is aligned with them, or with which of them (was it explicitly designed with these criteria in mind?). This is also important because Wilson et al. present and compare a range of evaluation methods for IAMs, so it would be necessary to state explicitly how and to what extent the tool contributes to the programme that Wilson et al. call for.
- The introduction also refers to the "mapmaker and navigator" approach to IAM-generated scenarios, which was central in the IPCC AR5. On this note, the quote from Beck and Oomen (line 20) should be introduced in a way that is less ambiguous, because as of now it suggests that the metaphor is Beck and Oomen's, while they are in fact explaining the vision outlined in Edenhofer and Minx (2014) and how it understands the relation between the COP and the IPCC; so it would be better to rephrase as "In this approach, the role of the COP is understood through a metaphor : 'the COP operates as a navigator...'" or something along this line.

This approach was also specific to AR5, and indeed was formulated by the co-chair of WGIII in that cycle. It would be relevant to mention developments in the AR6 cycle as well, especially the discussions about the design, scope and use of the Scenarios Database (which was arguably designed as the organising device for the use of scenarios in the report of WGIII, and informed by feedback on and criticism of the approach to IAM scenarios in AR5).

This seems especially relevant here because it would make it possible to discuss how *piamValidation* relates to other initiatives for harmonizing/comparing/validating IAM scenarios, including within the IAMC, in relations to Scenario Databases/explorers, and, more focused on the AR6, to the discussions about the way the vetting of scenarios was carried out in the AR6 and how to work with scenario ensembles (e.g. having a tool that can help compare across scenarios and compare scenarios to historical data could be presented as potential a contribution to future vetting processes). The following reference (published after the initial submission of the manuscript) maybe relevant for that discussion:

Sognnaes, I., Peters, G.P. Influence of individual models and studies on quantitative mitigation findings in the IPCC Sixth Assessment Report. *Nat Commun* **16**, 8343 (2025). https://doi.org/10.1038/s41467-025-64091-w

- *piamValidation* is presented as a tool that can improve the realism and reliability of IAMs on the one hand, and their transparency on the other hand. These are two often related but distinct requirements. The abstract is quite clear on this distinction, but the bulk of the paper is more focused on realism (with the application to REMIND as a clear example of how the tool can help improve realism) and the distinction and articulation between the two objectives (which can of course be combined) could be made even clearer. The NGFS application shows that the tool is more versatile and can have broader uses, oriented more towards intercomparison and handling of large scenario datasets, which is a strength that could be highlighted more. The value of scenarios is not only/not necessarily always in their realism – in some cases exploratory, idealized or extreme scenarios can be useful, and in that case a tool that can make comparison more robust and transparent is valuable; the capacity of the tool to adapt to a range of conceptions, approaches and purpose of scenarios development (instead of just driving harmonization/convergence towards one vision of realism or usefulness of scenarios) is then important.

- In terms of use for model improvement, judging from authors' affiliations, it seems that the tool was developed by the same team which develops REMIND, and so it is presumably particularly suited to REMIND. Would such use for model validation and improvement be replicable with other IAMs, i.e. would the tool be suitable and/or adaptable to other model architectures? This does not need to be tested for this paper, but the question would deserve to be opened in the conclusion.

- The question of the reference data, its availability, selection and processing is not discussed extensively. The conclusion states that "The REMIND case study demonstrates that the usefulness of the tool for historical and benchmark validation critically depends on the chosen validation settings and the availability of reliable reference data" but I did not find that this so was clearly explained in

the relevant section. The selection of reference data does raise some questions. Could the tool also be used to compare different sources of reference data? In the REMIND case study, it is explained that the tool was used to compare with "historical data" but the comparison extends to 2030 and includes projected trends. Why are IEA trends to 2030 deemed more reliable than REMIND near-terms trends and why calibrate against them? How does this relate to the ambition to improve transparency, considering that IEA data is not open and IEA models are less transparently documented than most IAMs?

- The conclusion (line 386) mentions "the ability to identify high-quality scenarios from an extensive scenario database". The reference to "high quality" begs the question of what defines the "quality" of a scenario: is it realism? fitness for purpose and ability to help address/clarify a specific question? replicability and transparency? Is the quality of scenario an absolute attribute or conditional on context and purpose of the scenario exercise? etc... To avoid opening these epistemological questions in the conclusion, it might be better to use a more neutral phrase e.g. "the ability to handle, compare and vet scenarios from an extensive scenario database".

---

## Author Response (AR2)

Response to Anonymous Referee #4

*Throughout the document, the original comments of the anonymous referee are presented sequentially in **black** and italic font.*

The authors' responses are provided in **blue** font to ensure clear distinction.

*Comment: The manuscript introduces and details the piamValidation package, developed to "enable systematic comparison of variables" from IAM datasets. The package is presented as an instrument that can improve reliability and transparency of IAMs, and two uses are highlighted: the comparison and harmonization of scenarios across IAMs, and the comparison of near-term dynamics in IAM scenarios with external, historical datasets. Two applications, corresponding to these two use cases, are then presented: one on NGFS scenarios, highlighting the different types of comparisons/harmonization that the tool can perform, and one for the improvement of the REMIND model through the validation of near-term technology dynamics.*

*The manuscript is clearly written and provides a good overview, description and discussion of the package. I recommend its publication.*

*Below are a few points that deserve clarification and/or discussion.*

**4 Response:** We are grateful to Reviewer #4 for the comprehensive and constructive feedback. The insightful remarks helped refine the manuscript and strengthen its clarity.

*Comment: Section 3.2 "REMIND Validate short-term technology trends" presents an application of the piamValidation package to REMIND scenario data and discusses subesquent changes between REMiND 3.2.0 and REMIND 3.5.0 to improve the accuracy of the near-term technological dynamics in the model, but it is not always clear how exactly the use of piamValidation informed the changes. This is especially the case in subsections 3.3 and 3.4 where the outcomes from the use of piamValidation are not mentioned explicitly, so it does not come out clearly how the changes implemented relate to the diagnosis (as opposed to updating/refining in the light of recent literature or data only). Section 3.5 is much clearer in that respect.*

**4.1 Response:** Thank you for this comment. In the second application case, the *piamValidation* tool is used to identify deviations from near-term trends and to document improvements that result from model adjustments. We have added clarifying remarks to the respective subsections which complement the general explanation on the validation and subsequent improvement process in section 3.2.

*Comment: Very minor point on this section, but for consistency the subsections should be labelled "3.2.1/3.2.2/3.2.3" rather than "3.3/3.4/3.5".*

**4.2 Response:** We agree and have adjusted the manuscript accordingly.

***Comment****: The introduction refers to "the key evaluation criterion outlined by Wilson et al. (2021)". It would need to briefly state what these criteria are and briefly explain how the piamValidation tool is aligned with them, or with which of them (was it explicitly designed with these criteria in mind?). This is also important because Wilson et al. present and compare a range of evaluation methods for IAMs, so it would be necessary to state explicitly how and to what extent the tool contributes to the programme that Wilson et al. call for.*

**4.3 Response:** Thank you for this important comment. We now mention the evaluation criteria by Wilson et al. (2021) and clarify how the *piamValidation* tool aligns with their framework. Specifically, we highlight the evaluation methods that the tool supports and thus can help to strengthen the evaluation criteria, with a particular focus on credibility.

***Comment****: The introduction also refers to the "mapmaker and navigator" approach to IAM-generated scenarios, which was central in the IPCC AR5. On this note, the quote from Beck and Oomen (line 20) should be introduced in a way that is less ambiguous, because as of now it suggests that the metaphor is Beck and Oomen's, while they are in fact explaining the vision outlined in Edenhofer and Minx (2014) and how it understands the relation between the COP and the IPCC; so it would be better to rephrase as "In this approach, the role of the COP is understood through a metaphor : 'the COP operates as a navigator…'" or something along this line.*

*This approach was also specific to AR5, and indeed was formulated by the co-chair of WGIII in that cycle. It would be relevant to mention developments in the AR6 cycle as well, especially the discussions about the design, scope and use of the Scenarios Database (which was arguably designed as the organising device for the use of scenarios in the report of WGIII, and informed by feedback on and criticism of the approach to IAM scenarios in AR5).*

*This seems especially relevant here because it would make it possible to discuss how piamValidation relates to other initiatives for harmonizing/comparing/validating IAM scenarios, including within the IAMC, in relations to Scenario Databases/explorers, and, more focused on the AR6, to the discussions about the way the vetting of scenarios was carried out in the AR6 and how to work with scenario ensembles (e.g. having a tool that can help compare across scenarios and compare scenarios to historical data could be presented as potential a contribution to future vetting processes and analyses of scenarios databases). The following reference (published after the initial submission of the manuscript) may be relevant for that discussion:*

*Sognnaes, I., Peters, G.P. Influence of individual models and studies on quantitative mitigation findings in the IPCC Sixth Assessment Report. Nat Commun 16, 8343 (2025). https://doi.org/10.1038/s41467-025-64091-w*

**4.4 Response:** We acknowledge that the original phrasing could have been interpreted ambiguously and have revised the manuscript accordingly. We also appreciate the reference to climate model validation in AR6 WGIII Chapter 3 and have incorporated this point into the revised manuscript.

*Comment: piamValidation is presented as a tool that can improve the realism and reliability of IAMs on the one hand, and their transparency on the other hand. These are two often related but distinct requirements. The abstract is quite clear on this distinction, but the bulk of the paper is more focused on realism (with the application to REMIND as a clear example of how the tool can help improve realism) and the distinction and articulation between the two objectives (which can of course be combined) could be made even clearer. The NGFS application shows that the tool is more versatile and can have broader uses, oriented more towards intercomparison and handling of large scenario datasets, which is a strength that could be highlighted more. The value of scenarios is not only/not necessarily always in their realism – in some cases exploratory, idealized or extreme scenarios can be useful, and in that case a tool that can make comparison more robust and transparent is valuable; the capacity of the tool to adapt to a range of conceptions, approaches and purpose of scenarios development (instead of just driving harmonization/convergence towards one vision of realism or usefulness of scenarios) is then important.*

**4.5 Response**: Thank you for this insightful comment. In Section 3.1, the first application case using NGFS scenarios illustrates different types of application cases of the *piamValidation* tool. This application case is not focused on near-term realism, but instead highlights the versatility of the tool. In contrast, Section 3.2 demonstrates the use of *piamValidation* tool to spot near-term deviations in the REMIND model and document subsequent improvements.

The observation that scenarios should be allowed to span a wide range of possible pathways instead of focusing on one most likely scenario is also the reason why we limit the validation checks to the near-term. Depending on technology, limits to scale-up, technology readiness and installation speeds allow the definition of feasibility boundaries which translates to the definition of thresholds up to 2030.

Finally, overall transparency is promoted through open distribution of configuration files and by supporting community-driven discussion on threshold choices and validation settings.

*Comment: In terms of use for model improvement, judging from authors' affiliations, it seems that the tool was developed by the same team which develops REMIND, and so it is presumably particularly suited to REMIND. Would such use for model validation and improvement be replicable with other IAMs, i.e. would the tool be suitable and/or adaptable to other model architectures? This does not need to be tested for this paper, but the question would deserve to be opened in the conclusion.*

**4.6 Response:** The model is agnostic in respect to which IAM is used and more generally could theoretically be applied to other types of models as well. The only restriction is on the structure of the data which is required to follow the IAMC format as described in section 2.2. This is also reflected in section 3.1 where the tool is applied to model output from MESSAGE and GCAM besides REMIND.

*Comment: The question of the reference data, its availability, selection and processing is not discussed extensively. The conclusion states that "The REMIND case study demonstrates that the usefulness of the tool for historical and benchmark validation critically depends on the chosen validation settings and the availability of reliable reference data" but I did not find that this so was clearly explained in the relevant section. The selection of reference data does raise some questions. Could the tool also be used to compare different sources of reference data?*

*In the REMIND case study, it is explained that the tool was used to compare with "historical data" but the comparison extends to 2030 and includes projected trends. Why are IEA trends to 2030 deemed more reliable than REMIND near-terms trends and why calibrate against them? How does this relate to the ambition to improve transparency, considering that IEA data is not open and IEA models are less transparently documented than most IAMs?*

**4.7 Response:** The selection of observational data and projected estimates is discussed in section 3.2 with the most detailed explanations described in the GitHub discussions linked in the manuscript (https://github.com/pik-piam/mrremind/discussions/544). We agree that care is needed when using other models to validate REMIND. We do think the comparison to specialized sector models with high granularity can yield valuable insights for a full-system model such as REMIND. (In the interest of transparency, open data and open-source models should be preferred where available.)

We see the presented validation exercise as a starting point of an ongoing community effort to identify reliable reference data and welcome discussions that help curate a collection of thresholds. We acknowledge uncertainty in external data sources by adding additional tolerances to reference values while also aiming to validate against the data range spanned by multiple sources where possible (see section 2.2.2).

The tool could also be used to compare different sources of reference data by treating them as different models and performing a model-intercomparison akin to the one presented in section 3.1.2.

*Comment: The conclusion (line 386) mentions "the ability to identify high-quality scenarios from an extensive scenario database". The reference to "high quality" begs the question of what defines the "quality" of a scenario: is it realism? fitness for purpose and ability to help address/clarify a specific question? replicability and transparency? Is the quality of scenario an absolute attribute or conditional on context and purpose of the scenario exercise? etc… To avoid opening these epistemological questions in the conclusion, it might be better to use a more neutral phrase e.g. "the ability to handle, compare and vet scenarios from an extensive scenario database".*

**4.8 Response:** Thank you for this comment. We agree and have changed the text accordingly.

Literature

Wilson, C., Guivarch, C., Kriegler, E., Van Ruijven, B., Van Vuuren, D.P., Krey, V., Schwanitz, V.J., Thompson, E.L., 2021. Evaluating process-based integrated assessment models of climate change mitigation. Climatic Change 166, 3. https://doi.org/10.1007/s10584-021-03099-9